# Statistical mechanics of low-rank tensor decomposition

**Jonathan Kadmon**
Department of Applied Physics, Stanford University
kadmonj@stanford.edu

**Surya Ganguli**
Department of Applied Physics, Stanford University and Google Brain, Mountain View, CA
sganguli@stanford.edu

## Abstract

Often, large, high dimensional datasets collected across multiple modalities can be organized as a higher order tensor. Low-rank tensor decomposition then arises as a powerful and widely used tool to discover simple low dimensional structures underlying such data. However, we currently lack a theoretical understanding of the algorithmic behavior of low-rank tensor decompositions. We derive Bayesian approximate message passing (AMP) algorithms for recovering arbitrarily shaped low-rank tensors buried within noise, and we employ dynamic mean field theory to precisely characterize their performance. Our theory reveals the existence of phase transitions between easy, hard and impossible inference regimes, and displays an excellent match with simulations. Moreover it reveals several qualitative surprises compared to the behavior of symmetric, cubic tensor decomposition. Finally, we compare our AMP algorithm to the most commonly used algorithm, alternating least squares (ALS), and demonstrate that AMP significantly outperforms ALS in the presence of noise.

## 1 Introduction

The ability to take noisy, complex data structures and decompose them into smaller, interpretable components in an unsupervised manner is essential to many fields, from machine learning and signal processing [1, 2] to neuroscience [3]. In datasets that can be organized as an order 2 data matrix, many popular unsupervised structure discovery algorithms, like PCA, ICA, SVD or other spectral methods, can be unified under rubric of low rank matrix decomposition. More complex data consisting of measurements across multiple modalities can be organized as higher dimensional data arrays, or higher order tensors. Often, one can find simple structures in such data by approximating the data tensor as a sum of rank 1 tensors. Such decompositions are known by the name of rank-decomposition, CANDECOMP/PARAFAC or CP decomposition (see [4] for an extensive review).

The most widely used algorithm to perform rank decomposition is alternating least squares (ALS) [5, 6], which uses convex optimization techniques on different slices of the tensor. However, a major disadvantage of ALS is that it does not perform well in the presence of highly noisy measurements. Moreover, its theoretical properties are not well understood. Here we derive and analyze an approximate message passing (AMP) algorithm for optimal Bayesian recovery of arbitrarily shaped, high-order low-rank tensors buried in noise. As a result, we obtain an AMP algorithm that both out-performs ALS and admits an analytic theory of its performance limits.

AMP algorithms have a long history dating back to early work on the statistical physics of perceptron learning [7, 8] (see [9] for a review). The term AMP was coined by Donoho, Maleki and Montanari in their work on compressed sensing [10] (see also [11, 12, 13, 14, 15, 16] for replica approaches to compressed sensing and high dimensional regression). AMP approximates belief propagation in graphical models and a rigorous analysis of AMP was carried out in [17]. For a rank-one matrix estimation problem, AMP was first introduced and analyzed in [18]. This framework has been extended in a beautiful body of work by Krzakla and Zdeborova and collaborators to various low-rank matrix factorization problems in [19, 20, 21, 22]). Recently low-rank tensor decomposition through AMP was studied in [21], but their analysis was limited to symmetric tensors which are then necessarily cubic in shape. In [23], a similar approach was used to extend the analysis of order-2 tensors (matrices) to order-3 tensors, which can potentially be further extended to higher orders.

However, tensors that occur naturally in the wild are almost never cubic in shape, nor are they symmetric. The reason is that the $p$ different modes of an order $p$ tensor correspond to measurements across very different modalities, resulting in very different numbers of dimensions across modes, yielding highly irregularly shaped, non-cubic tensors with no symmetry properties. For example in EEG studies 3 different tensor modes could correspond to time, spatial scale, and electrodes [24]. In fMRI studies the modes could span channels, time, and patients [25]. In neurophysiological measurements they could span neurons, time, and conditions [26] or neurons, time, and trials [3]. In studies of visual cortex, modes could span neurons, time and stimuli [27].

Thus, given that tensors in the wild are almost never cubic, nor symmetric, to bridge the gap between theory and experiment, we go beyond prior work to derive and analyze Bayes optimal AMP algorithms for *arbitrarily shaped* high order and low rank tensor decomposition with *different* priors for different tensor modes, reflecting their different measurement types. We find that the low-rank decomposition problem admits two phase transitions separating three qualitatively different inference regimes: (1) the easy regime at low noise where AMP works, (2) the hard regime at intermediate noise where AMP fails but the ground truth tensor is still possible to recover, if not in a computationally tractable manner, and (3) the impossible regime at high noise where it is believed no algorithm can recover the ground-truth low rank tensor.

From a theoretical perspective, our analysis reveals several surprises relative to the analysis of symmetric cubic tensors in [21]. First, for symmetric tensors, it was shown that the easy inference regime *cannot* exist, *unless* the prior over the low rank factor has non-zero mean. In contrast, for non-symmetric tensors, one tensor mode *can* have zero mean *without* destroying the existence of the easy regime, as long as the other modes have non-zero mean. Furthermore, we find that in the space of all possible tensor shapes, the hard regime has the largest width along the noise axis when the shape is cubic, thereby indicating that tensor shape can have a strong effect on inference performance, and that cubic tensors have highly non-generic properties in the space of all possible tensor shapes.

Before continuing, we note some connections to the statistical mechanics literature. Indeed, AMP is closely equivalent to the TAP equations and the cavity method [28, 29] in glassy spin systems. Furthermore, the posterior distribution of noisy tensor factorization is equivalent to $p$-spin magnetic systems [30], as we show below in section 2.2. For Bayes-optimal inference, the phase space of the problem is reduced to the Nishimori line [31]. This ensures that the system does not exhibit replica-symmetry breaking. Working in the Bayes-optimal setting thus significantly simplifies the statistical analysis of the model. Furthermore, it allows theoretical insights into the inference phase-transitions, as we shall see below. In practice, for many applications the prior or underlying rank of the tensors are not known *a-priori*. The algorithms we present here can also be applied in a non Bayes-optimal setting, where the parametric from of the prior can not be determined. In that case, the theoretical asymptotics we describe here may not hold. However, approximate Bayesian-optimal settings can be recovered through parameter learning using expectation-maximization algorithms [32]. We discuss these consequences in section 4. Importantly,the connection to the statistical physics of magnetic systems allows the adaptation of many tools and intuitions developed extensively in the past few decades, see e.g. [33]. We discuss more connections to statistical mechanics as we proceed below.

## 2 Low rank decomposition using approximate message passing

In the following we define the low-rank tensor decomposition problem and present a derivation of AMP algorithms designed to solve this problem, as well as a dynamical mean field theory analysis of their performance. A full account of the derivations can be found in the supplementary material.

### 2.1 Low-rank tensor decomposition

Consider a general tensor $Y$ of order-$p$, whose components are given by a set of $p$ indices, $Y_{i_1,i_2,\ldots,i_p}$. Each index $i_\alpha$ is associated with a specific *mode* of the tensor. The dimension of the mode $\alpha$ is $N_\alpha$ so the index $i_\alpha$ ranges from $1,\ldots,N_\alpha$. If $N_\alpha = N$ for all $\alpha$ then the tensor is said to be *cubic*. Otherwise we define $N$ as the geometric mean of all dimensions $N = (\prod_\alpha^p N_\alpha)^{1/p}$, and denote $n_\alpha \equiv N_\alpha/N$ so that $\prod_\alpha^p n_\alpha = 1$. We employ the shorthand notation $Y_{i_1,i_2,\ldots,i_p} \equiv Y_a$, where $a = \{i_1,\ldots,i_p\}$ is a set of $p$ numbers indicating a specific element of $Y$. A rank-1 tensor of order-$p$ is the outer product of $p$ vectors (order-1 tensors) $\prod_{1\leq\alpha\leq p}^{\otimes} \mathbf{x}_\alpha$,where $\mathbf{x}_\alpha \in \mathbb{R}^{N_\alpha}$. A rank-$r$ tensor of order-$p$ has a special structure that allows it to be decomposed into a sum of $r$ rank-1 tensors, each of order-$p$. The goal of the rank decomposition is to find all $\mathbf{x}_\alpha^\rho \in \mathbb{R}^{N_\alpha}$ , for $\alpha = 1,..p$, and $\rho = 1,\ldots,r$, given a tensor $Y$ of order-$p$ and rank-$r$. In the following, we will use $\mathbf{x}_{\alpha i} \in \mathbb{R}^r$ to denote the vector of values at each entry of the tensor, spanning the $r$ rank-1 components. In a low-rank decomposition it is assumed that $r < N$. In *noisy* low-rank decomposition, individual elements $Y_a$ are noisy measurements of a low-rank tensor [Figure 1.A]. A comprehensive review on tensor decomposition can be found in [4].

We state the problem of low-rank noisy tensor decomposition as follows: Given a rank-$r$ tensor

$$w_a = \frac{1}{N^{\frac{p-1}{2}}} \sum_{\rho=1}^r \prod_\alpha x_{\alpha i}^\rho, \tag{1}$$

we would like to find all the underlying factors $x_{\alpha i}^\rho$. We note that we have used the shorthand notation $i = i_\alpha$ to refer to the index $i_\alpha$ which ranges from 1 to $N_\alpha$, i.e. the dimensionality of mode $\alpha$ of the tensor.

Now consider a noisy measurement of the rank-$r$ tensor $w$ given by

$$Y = w + \sqrt{\Delta}\epsilon, \tag{2}$$

where $\epsilon$ is a random noise tensor of the same shape as $w$ whose elements are distributed i.i.d according to a standard normal distribution, yielding a total noise variance $\Delta \sim O(1)$ [Fig. 1.A]. The underlying factors $x_{\alpha i}^\rho$ are sampled i.i.d from a *prior* distribution $P_\alpha(x)$, that may vary between the modes $\alpha$. This model is a generalization of the spiked-tensor models studied in [21, 34].

We study the problem in the thermodynamic limit where $N \to \infty$ while $r, n_\alpha \sim O(1)$. In that limit, the mean-field theory we derive below becomes exact. The achievable performance in the decomposition problem depends on the signal-to-noise ratio (SNR) between the underlying low rank tensor (the signal) and the noise variance $\Delta$. In eq. (1) we have scaled the SNR (signal variance divided by noise variance) with $N$ so that the SNR is proportional to the ratio between the $O(N)$ unknowns and the $N^p$ measurements, making the inference problem neither trivially easy nor always impossible. From a statistical physics perspective, this same scaling ensures that the posterior distribution over the factors given the data corresponds to a Boltzmann distribution whose Hamiltonian has extensive energy proportional to $N$, which is necessary for nontrivial phase transitions to occur.

### 2.2 Tensor decomposition as a Bayesian inference problem

In Bayesian inference, one wants to compute properties of the posterior distribution

$$P(w|Y) = \frac{1}{Z(Y,w)} \prod_\rho^r \prod_\alpha^p \prod_i^N P_\alpha(x_{\alpha i}^\rho) \prod_a P_{out}(Y_a|w_a). \tag{3}$$

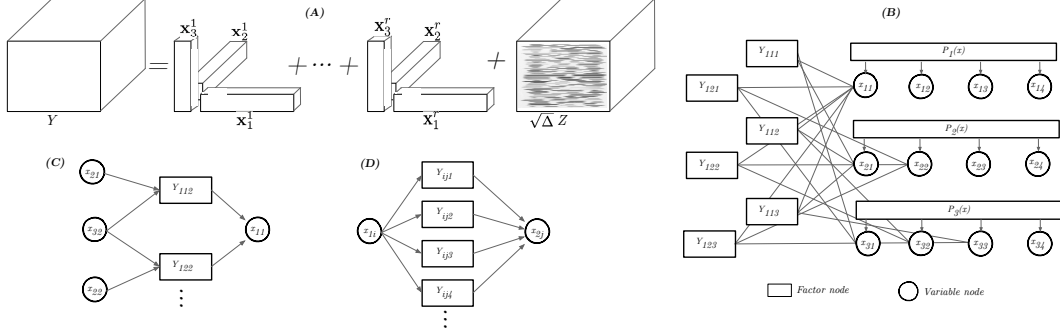

Figure 1: *Low rank-decomposition of an order-3 spiked-tensor.* **(A)** The observation tensor $Y$ is a sum of $r$ $rank - 1$ tensors and a noise tensor $\epsilon$ with variance $\Delta$. **(B)** Factor graph for the decomposition of an order-3 tensor. **(C)** Incoming messages into each variable node arrives from variable nodes connected to the adjacent factor nodes. **(D)** Each node receives $N^{p-2}$ messages from each of the other variable nodes in the graph.

Here $P_{out}\left(Y_a \,|w_a\right)$ is an element-wise output channel that introduce independent noise into individual measurements. For additive white Gaussian noise, the output channel in eq. (3) is given by

$$\log P_{out}\left(Y_a \,|w_a\right) = g\left(Y_a \,|w_a\right) = \frac{1}{2\Delta}\left(Y_a - w_a\right)^2 - \frac{1}{2}\log 2\pi\Delta, \tag{4}$$

where $g(\cdot)$ is a quadratic *cost function*. The denominator $Z(Y, w)$, in 3 is a normalization factor, or the partition function in statistical physics. In a Bayes-optimal setting, the priors $P_\alpha(x)$, as well as the rank $r$ and the noise $\Delta$ are known.

The channel universality property [19] states that for low-rank decomposition problems, $r \ll N$, any output channel is equivalent to simple additive white Gaussian noise, as defined in eq. (2). Briefly, the output channel can be developed as a power series in $w_a$. For low-rank estimation problems we have $w_a \ll 1$ [eq. (1)], and we can keep only the leading terms in the expansion. One can show that the remaining terms are equivalent to random Gaussian noise, with variance equal to the inverse of the Fisher information of the channel [See supplementary material for further details]. Thus, non-additive and non-Gaussian measurement noise at the level of individual elements, can be replaced with an effective additive Gaussian noise, making the theory developed here much more generally applicable to diverse noise scenarios.

The motivation behind the analysis below, is the observation that the posterior (3), with the quadratic cost function (4) is equivalent to a Boltzmann distribution of a magnetic system at equilibrium, where $\mathbf{x}_{\alpha i} \in \mathbb{R}^r$ can be though of as the $r$-dimensional vectors of a spherical (xy)-spin model [30].

## 2.3 Approximate message passing on factor graphs

To solve the problem of low-rank decomposition we frame the problem as a graphical model with an underlying bipartite factor graph. The variable nodes in the graph represent the $rN \sum_\alpha n_\alpha$ unknowns $x_{\alpha i}^\rho$ and the $N^p$ factor nodes correspond to the measurements $Y_a$. The edges in the graph are between factor node $Y_a$ and the variable nodes in the neighbourhood $\partial a$ [Figure 1.B]. More precisely, for each factor node $a = \{i_1, i_2, ..., i_p\}$, the set of variable nodes in the neighbourhood $\partial a$ are precisely $\{\boldsymbol{x}_{1i_1}, \boldsymbol{x}_{2i_2}, ..., \boldsymbol{x}_{pi_p}\}$, where each $\boldsymbol{x}_{\alpha i_\alpha} \in \mathbb{R}^r$. Again, in the following we will use the shorthand notation $\boldsymbol{x}_{\alpha i}$ for $\boldsymbol{x}_{\alpha i_\alpha}$. The state of a variable node is defined as the marginal probability distribution $\eta_{\alpha i}(\mathbf{x})$ for each of the $r$ components of the vectors $\mathbf{x}_{\alpha i} \in \mathbb{R}^r$. The estimators $\hat{\mathbf{x}}_{\alpha i} \in \mathbb{R}^r$ for the values of the factors $\mathbf{x}_{\alpha i}$ are given by the means of each of the marginal distributions $\eta_{\alpha i}(\mathbf{x})$.

In the approximate message passing framework, the state of each node (also known as a 'belief'), $\eta_{\alpha i}(x)$ is transmitted to all other variable nodes via its adjacent factor nodes [Fig. 1.C]. The state of each node is then updated by marginalizing over all the incoming messages, weighted by the cost function and observations in the factor nodes they passed on the way in:

$$\eta_{\alpha i}(\mathbf{x}) = \frac{P_\alpha(\mathbf{x})}{Z_{\alpha i}} \prod_{a \in \partial \alpha i} \prod_{\beta j \in \partial a \backslash \alpha i} Tr_{x_{\beta j}} \eta_{\beta j}(x_{\beta j}) e^{g(y_a, w_a)}. \tag{5}$$

Here $P_\alpha(\mathbf{x})$ is the prior for each factor $\mathbf{x}_{\alpha i}$ associated with mode $\alpha$, and $Z_{\alpha i} = \int d\mathbf{x}\eta_{\alpha i}(\mathbf{x})$ is the partition function for normalization. The first product in (5) spans all factor nodes adjacent to variable node $\alpha i$. The second product is over all variable nodes adjacent to each of the factor nodes, excluding the target node $\alpha i$. The trace $Tr_{x_{\beta j}}$ denotes the marginalization of the cost function $g(y_a, w_a)$ over all incoming distributions.

The mean of the marginalized posterior at node $\alpha i$ is given by

$$\hat{\mathbf{x}}_{\alpha i} = \int dx \eta_{\alpha i}(\mathbf{x})\mathbf{x} \in \mathbb{R}^r, \tag{6}$$

and its covariance is

$$\hat{\sigma}^2_{\alpha i} = \int dx \eta_{\alpha i}(x)xx^T - \hat{x}_{\alpha i}\hat{x}^T_{\alpha i} \in \mathbb{R}^{r \times r}. \tag{7}$$

Eq. (5) defines an iterative process for updating the beliefs in the network. In what follows, we use mean-field arguments to derive iterative equations for the means and covariances of the these beliefs in (6)-(7). This is possible given the assumption that incoming messages into each node are probabilistically independent. Independence is a good assumption when short loops in the underlying graphical model can be neglected. One way this can occur is if the factor graph is sparse [35, 36]. Such graphs can be approximated by a directed acyclic graph; in statistical physics this is known as the Bethe approximation [37]. Alternatively, in low-rank tensor decomposition, the statistical independence of incoming messages originates from weak pairwise interactions that scale as $w \sim N^{-(p-1)/2}$. Loops correspond to higher order terms interaction terms, which become negligible in the thermodynamic limit [17, 33].

Exploiting these weak interactions we construct an accurate mean-field theory for AMP. Each node $\alpha i$ receives $N^{p-2}$ messages from every node $\beta j$ with $\beta \neq \alpha$, through all the factor nodes that are connected to both nodes,$\{y_b|b \in \partial\alpha i \cup \partial\beta j\}$ [Fig. 1.D]. Under the independence assumption of incoming messages, we can use the central limit theorem to express the state of node $\alpha j$ in (5) as

$$\eta_{\alpha i}(\mathbf{x}) = \frac{P_\alpha(\mathbf{x})}{Z_\alpha(A_{\alpha i}, \mathbf{u}_{\alpha i})} \prod_{\beta j \neq \alpha i} \exp\left(-\mathbf{x}^T A_{\beta j}\mathbf{x} + \mathbf{u}^T_{\beta j}\mathbf{x}\right), \tag{8}$$

where $A^{-1}_\beta \mathbf{u}_{\beta j}$ and $A^{-1}_\beta$ are the mean and covariance of the local incoming messages respectively. The distribution is normalized by the partition function

$$Z_\alpha(A, \mathbf{u}) = \int dx P_\alpha(x) \exp\left[\left(\mathbf{u}^T\mathbf{x} - \mathbf{x}^T A\mathbf{x}\right)\right]. \tag{9}$$

The mean and covariance of the distribution, eq. (6) and (7) are the moments of the partition function

$$\hat{\mathbf{x}}_{\alpha i} = \frac{\partial}{\partial \mathbf{u}_{\alpha i}} \log Z_\alpha, \ \hat{\sigma}_{\alpha i} = \frac{\partial^2}{\partial \mathbf{u}_{\alpha i}\partial \mathbf{u}^T_{\alpha i}} \log Z_\alpha. \tag{10}$$

Finally, by expanding $g(Y_a, w_a)$ in eq. (5) to quadratic order in $w$, and averaging over the posterior, one can find a self consistent equation for $A_{\alpha i}$ and $\mathbf{u}_{\alpha i}$ in terms of $\mathbf{x}_{\alpha i}$ and $Y$ [see supplemental material for details].

## 2.4 AMP algorithms

Using equations (10), and the self-consistent equations for $A_{\alpha i}$ and $\mathbf{u}_{\alpha i}$, we construct an iterative algorithm whose dynamics converges to the solution of the self-consistent equations [ see supplemental

material for details]. The resulting update equations for the parameters are

$$\mathbf{u}_{\alpha i}^t = \frac{n_\alpha}{\Delta N^{(p-1)/2}} \sum_{a \in \partial \alpha i} Y_a \left( \prod_{(\beta,j) \in \partial b \backslash (\alpha,i)}^{\odot} \hat{\mathbf{x}}_{\beta j}^t \right) - \frac{1}{\Delta} \hat{\mathbf{x}}_{\alpha i}^{t-1} \sum_{\beta \neq \alpha} \Sigma_\beta^t \odot D_{\alpha\beta}^{t,t-1} \qquad (11)$$

$$A_\alpha^t = \frac{1}{\Delta} \prod_{\beta \neq \alpha}^{\odot} \left( \frac{1}{n_\beta N} \sum_{j=1}^{N} \hat{\mathbf{x}}_{\beta j}^t \hat{\mathbf{x}}_{\beta j}^{tT} \right) \qquad (12)$$

$$\hat{\mathbf{x}}_{\alpha i}^{t+1} = \frac{\partial}{\partial \mathbf{u}_{\alpha i}^t} \log Z_\alpha(A_\alpha^t, \mathbf{u}_{\alpha i}^t) \qquad (13)$$

$$\hat{\sigma}_{\alpha i}^{t+1} = \frac{\partial^2}{\partial \mathbf{u}_{\alpha i}^t \partial \mathbf{u}_{\alpha i}^{tT}} \log Z_\alpha(A_\alpha^t, \mathbf{u}_{\alpha i}^t), \qquad (14)$$

The second term on the RHS of (11), is given by

$$D_{\alpha\beta}^{t,t-1} = \prod_{\gamma \neq \alpha, \beta}^{\odot} \left( \frac{1}{N} \sum_k \hat{\mathbf{x}}_{\gamma k}^t \hat{\mathbf{x}}_{\gamma k}^{t-1,T} \right), \quad \Sigma_\alpha^t = N^{-1} \sum_i \hat{\sigma}_{\alpha i}^t. \qquad (15)$$

This term originates from the the exclusion the target node $\alpha i$ from the product in equations (5) and (8). In statistical physics it corresponds to an Onsager reaction term due to the removal of the node yielding a cavity field [38]. In the above, the notations $\odot$, $\prod^{\odot}$ denote component-wise multiplication between two, and multiple tensors respectively.

Note that in the derivation of the iterative update equation above, we have implicitly used the assumption that we are in the Bayes-optimal regime which simplifies eq. (11)-(14) [see supplementary material for details]. The AMP algorithms can be derived without the assumption of Bayes-optimality, resulting in a slightly more complicated set of algorithms [See supplementary material for details]. However, further analytic analysis, which is the focus of this current work, and the derivation of the dynamic mean-field theory which we present below is applicable in the Bayes-optimal regime, were there is no replica-symmetry breaking, and the estimators are self-averaging. Once the update equations converge, the estimates for the factors $\mathbf{x}_{\alpha i}$ and their covariances are given by the fixed point value of equations (13) and (14) respectively. A statistical treatment for the convergence in typical settings is presented in the following section.

## 2.5 Dynamic mean-field theory

To study the performance of the algorithm defined by eq. (11)-(14), we use another mean-field approximation that estimates the evolution of the inference error. As before, the mean-field becomes exact in the thermodynamic limit. We begin by defining order parameters that measure the correlation of the estimators $\hat{\mathbf{x}}_{\alpha i}^t$ with the ground truth values $\mathbf{x}_{\alpha i}$ for each mode $\alpha$ of the tensor

$$M_\alpha^t = (n_\alpha N)^{-1} \sum_{i=1}^{N_\alpha} \hat{\mathbf{x}}_{\alpha i}^t \mathbf{x}_{\alpha i}^T \in \mathbb{R}^{r \times r}. \qquad (16)$$

Technically, the algorithm is permutation invariant, so one should not expect the high correlation values to necessarily appear on the diagonal of $M_\alpha^t$. In the following, we derive an update equation for $M_\alpha^t$, which will describe the performance of the algorithm across iterations.

An important property of Bayes-optimal inference is that there is no statistical difference between functions operating on the ground truth values, or on values sampled uniformly from the posterior distribution. In statistical physics this property is known as one of the *Nishimori conditions* [31]. These conditions allow us to derive a simple equation for the update of the order parameter (16). For example, from (12) one easily finds that in Bayes-optimal settings $A_\alpha^t = \bar{M}_\alpha^t$. Furthermore, averaging the expression for $\mathbf{u}_{\alpha i}$ over the posterior, we find that [ supplemental material]

$$\mathbb{E}_{P(W|Y)} \left[ \mathbf{u}_{\alpha i}^t \right] = \bar{M}_\alpha^t x_{\alpha i}, \qquad (17)$$

where

$$\bar{M}_\alpha^t \equiv \frac{n_\alpha}{\Delta} \prod_{\beta \neq \alpha}^{\odot} M_\beta^t. \qquad (18)$$

Similarly, the covariance matrix of $\mathbf{u}_{\alpha i}$ under the posterior is

$$COV_{P(W|Y)}\left[\mathbf{u}_{i\alpha}^t\right] = \bar{M}_\alpha^t. \tag{19}$$

Finally, using eq. (13) for the estimation of $\hat{\mathbf{x}}_{\alpha i}$, and the definition of $M_\alpha^t$ in (16) we find a dynamical equation for the evolution of the order parameters $M_\alpha^t$:

$$M_\alpha^{t+1} = \mathbb{E}_{P_\alpha(\mathbf{x}),z}\left[f_\alpha\left(\bar{M}_\alpha^t, \bar{M}_\alpha^t x_{\alpha i} + \sqrt{\bar{M}_\alpha^t}\mathbf{z}\right)x_{\alpha i}^T\right], \tag{20}$$

where $f_\alpha \equiv \frac{\partial}{\partial \mathbf{u}}\log Z_\alpha(A, \mathbf{u})$ is the estimation of $\hat{\mathbf{x}}_{\alpha i}^{t+1}$ from (13). The average in (20) is over the prior $P_\alpha(\mathbf{x})$ and over the standard Gaussian variables $\mathbf{z} \in \mathbb{R}^r$. The average over $\mathbf{z}$ represents fluctuations in the mean $\bar{M}_\alpha^t x_{\alpha i}$ in (17), due to the covariance $\bar{M}_\alpha^t$ in (19).

Finally, the performance of the algorithm is given by the fixed point of (20),

$$M_\alpha^* = \mathbb{E}_{P_\alpha(\mathbf{x}),z}\left[f_\alpha\left(\bar{M}_\alpha^*, \bar{M}_\alpha^* x_{\alpha i} + \sqrt{\bar{M}_\alpha^*}\mathbf{z}\right)x_{\alpha i}^T\right], \text{ where } \bar{M}_\alpha^* \equiv \frac{n_\alpha}{\Delta}\prod_{\beta \neq \alpha}^{\odot} M_\beta^*. \tag{21}$$

As we will see below, the inference error can be calculated from the fixed point order parameters $M_\alpha^*$ in a straightforward manner.

## 3 Phase transitions in generic low-rank tensor decomposition

The dynamics of $M_\alpha^t$ depend on the SNR via the noise level $\Delta$. To study this dependence, we solve equations (20) and (21) with specific priors. Below we present the solution of using Gaussian priors. In the supplementary material we also solve for Bernoulli and Gauss-Bernoulli distributions, and discuss mixed cases where each mode of the tensor is sampled from a different prior. Given our choice of scaling in (1), we expect phase transitions at $O(1)$ values of $\Delta$, separating three regimes where inference is: (1) easy at small $\Delta$; (2) hard at intermediate $\Delta$; and (3) impossible at large $\Delta$. For simplicity we focus on the case of rank $r = 1$, where the order parameters $M_\alpha^t$ in (16) become scalars, which we denote $m_\alpha^t$.

### 3.1 Solution with Gaussian priors

We study the case where $x_{\alpha i}$ are sampled from normal distributions with mode-dependent mean and variance $P_\alpha(x) \sim \mathcal{N}(\mu_\alpha, \sigma_\alpha^2)$. The mean-field update equation (20) can be written as

$$m_\alpha^{t+1} = \frac{\frac{\mu_\alpha^2}{\sigma_\alpha^2} + \left(\sigma^2 + \mu^2\right)\bar{m}_\alpha^t}{\sigma^{-2} + \bar{m}_\alpha^t}, \tag{22}$$

where $\bar{m}_\alpha^t \equiv \Delta^{-1}n_\alpha\prod_{\beta \neq \alpha}m_\beta$ , as in (18). We define the average inference error for all modes

$$MSE = \frac{1}{p}\sum_\alpha\frac{|\hat{x}_\alpha - x_\alpha|^2}{2\sigma_\alpha^2} = \frac{1}{p}\sum_\alpha\left(1 + \frac{\mu_\alpha^2}{\sigma_\alpha^2} - \frac{1}{\sigma_\alpha^2}m_\alpha^*\right), \tag{23}$$

where $m_\alpha^*$ is the fixed point of eq. (22). Though we focus here on the $r = 1$ case for simplicity, the theory is equally applicable to higher-rank tensors.

Solutions to the theory in (22) and (23) are plotted in Fig. 2.A together with numerical simulations of the algorithm (11)-(14) for order-3 tensors generated randomly according to (2). The theory and simulations match perfectly. The AMP dynamics for general tensor decomposition is qualitatively similar to that of rank-1 symmetric matrix and tensor decompositions, despite the fact that such symmetric objects possess only one mode. As a consequence, the space of order parameters for these two problems is only one-dimensional; in contrast for the general case we consider here, it is $p$-dimensional. Indeed, the $p = 3$ order parameters are all simultaneously and correctly predicted by our theory.

For low levels of noise, the iterative dynamics converge to a stable fixed point of (22) with low MSE. As the noise increases beyond a bifurcation point $\Delta_{alg}$, a second stable fixed point emerges with

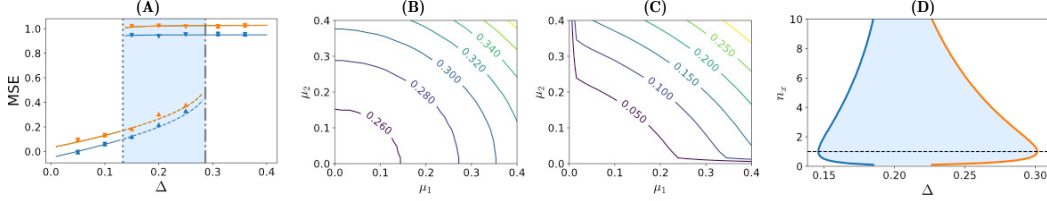

Figure 2: *Phase transitions in the inference of arbitrary order-3 rank-1 tensors.* **(A)** MSE at the fixed point plotted against the noise level $\Delta$. Shaded blue area marks the bi-stability regime $\Delta_{alg} < \Delta < \Delta_{dyn}$. Solid (dashed) lines show theoretically predicted MSE for random (informed) initializations. Points are obtained from numerical simulations [$\sigma_\alpha = 1$, $\mu_1 = \mu_2 = 0.1$ (blue), $\mu_3 = 0.3$ (orange), $N = 500$, $n_\alpha = 1$]. **(B)** Contours of $\Delta_{dyn}$ as a function of the two non-zero means of modes $\alpha = 1, 2$ [$\mu_3 = 0$, $\sigma_\alpha = 1$]. As either of the two nonzero means increases, $\Delta_{dyn}$ increases with them, reflecting an increase in the regime of noise level $\Delta$ over which inference is easy. Importantly, the transition $\Delta_{dyn}$ is finite even when only one prior has non zero mean. **(C)** Same as (B) but for $\Delta_{alg}$. Again, as either mean increases, $\Delta_{alg}$ increases also, reflecting a delay in the onset of the impossible regime as the noise level $\Delta$ increases. The algorithmic phase transition is finite when at most one prior has zero mean. **(D)** Lower and higher transition points $\Delta_{Alg}$ (blue) and $\Delta_{Dyn}$ (orange) as a function of tensor *shape*. The ratios between the mode dimensions are $n_\alpha = \{1, n_x, 1/n_x\}$. The width of the bi-stable or hard inference regime is widest at the cubic point where $n_x = 1$.

$m_\alpha^* \ll 1$ and $MSE \approx 1$. Above this point AMP may not converge to the true factors. The basin of attraction of the two stable fixed points are separated by a $p - 1$ dimensional sub-manifold in the $p$-dimensional order parameter space of $m_\alpha$. If the initial values $x_{\alpha i}^0$ have sufficiently high overlap with the true factors $x_{\alpha i}$, then the AMP dynamics will converge to the low error fixed point; we refer to this as the informative initialization, as it requires prior knowledge about the true structure. For uninformative initializations, the dynamics will converge to the high error fixed point almost surely in the thermodynamic limit.

At a higher level of noise, $\Delta_{dyn}$, another pitchfork bifurcation occurs and the high error fixed point becomes the *only* stable point. With noise levels $\Delta$ above $\Delta_{dyn}$, the dynamic mean field equations will always converge to a high error fixed point. In this regime AMP cannot overcome the high noise and inference is impossible.

From eq. (22), it can be easily checked that if the prior means $\mu_\alpha = 0$, $\forall \alpha$ then the high error fixed point with $m_\alpha = 0$ is *stable* for any finite $\Delta$. This implies that $\Delta_{alg} = 0$, and there is no easy regime for inference, so $AMP$ with uninformed initialization will never find the true solution. This difficulty was previously noted for the low-rank decomposition of symmetric tensors [21], and it was further shown there that the prior *must* be non-zero for the existence of an easy inference regime. However, for general tensors there is higher flexibility; one mode $\alpha$ *can* have a zero mean without destroying the existence of an easy regime. To show this we solved (22), with different prior means for different modes and we plot the phase boundaries in Fig. 2.B-C. For the $p = 3$ case, $\Delta_{dyn}$ is finite even if two of the priors have zero mean. Interestingly, the algorithmic transition $\Delta_{alg}$ is finite if at most one prior is has zero mean. Thus, the general tensor decomposition case is qualitatively different than the symmetric case in that an easy regime can exist even when a tensor mode has zero mean.

### 3.2 Non-cubic tensors

The shape of the tensor, defined by the different mode dimensions $n_\alpha$, has an interesting effect on the phase transition boundaries, which can be studied using (22). In figure 2.D the two transitions, $\Delta_{alg}$ and $\Delta_{dyn}$ are plotted as a function of the shape of the tensor. Over the space of all possible tensor shapes, the boundary between the hard and impossible regimes, $\Delta_{dyn}$ is maximized, or pushed furthest to the right in Fig. 2.D, when the shape takes the special cubic form where all dimensions are equal $n_\alpha = 1$, $\forall \alpha$. This diminished size of the impossible regime at the cubic point can be understood by noting the cubic tensor has the highest ratio between the number of observed data points $N^p$ and the number of unknowns $rN \sum_\alpha n_\alpha$.

Interestingly the algorithmic transition is lowest at this point. This means that although the ratio of observations to unknowns is the highest, algorithms may not converge, as the width of the hard

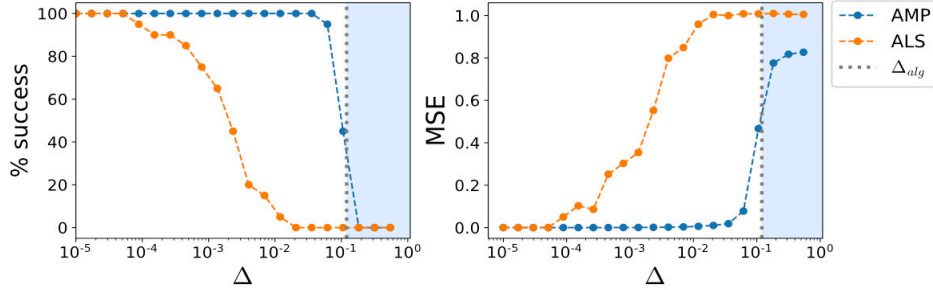

Figure 3: *Comparing AMP and ALS* . **Left**: The percentage out of 50 simulations that converged to the low error solution as a function of the noise $\Delta$ **Right**: MSE averaged over the 50 simulations. In both figures, the vertical dashed line is the theoretical predication of the algorithmic transition point $\Delta_{alg}$. [$p = 3$, $r = 1$, $\sigma_\alpha = 1$, $\mu_\alpha = 0.2$, $n_\alpha = \{1, \frac{8}{10}, \frac{10}{8}\}$, $N = 500$ ]

regime is maximized. To explain this observation, we note that in (18), the noise can be rescaled independently in each mode by defining $\Delta \to \Delta_\alpha = \Delta/n_\alpha$. It follows that for non-cubic tensors the worst case effective noise across modes will be necessarily higher than in the cubic case. As a consequence, moving from cubic to non-cubic tensors lowers the minimum noise level $\Delta_{alg}$ at which the uninformative solution is stable, thereby extending the hard regime to the left in Fig. 2.D.

# 4    Bayesian AMP compared to maximum a-posteriori (MAP) methods

We now compare the performance of AMP to one of the most commonly used algorithms in practice, namely alternating least squares (ALS) [4]. ALS is motivated by the observation that optimizing one mode while holding the rest fixed is a simple least-squares subproblem [6, 5]. Typically, ALS performs well at low noise levels, but here we explore how well it compares to AMP at high noise levels, in the scaling regime defined by defined by (1) and (2), where inference can be non-trivial.

In Fig. 3 we compare the performance of ALS with that of AMP on the same underlying large ($N = 500$) tensors with varying amounts of noise. First, we note that that ALS does not exhibit a sharp phase transition, but rather a smooth cross-over between solvable and unsolvable regimes. Second, the robustness of ALS to noise is much lower than that of AMP. This difference is more substantial as the size of the tensors, $N$, is increased [data not shown].

One can understand the difference in performance by noting that ALS is like a MAP estimator, while Bayesian AMP attempts to find the minimal mean square error (MMSE) solution. AMP does so by marginalizing probabilities at every node. Thus AMP is expected to produce better inferences when the posterior distribution is rough and dominated by noise. From a statistical physics perspective, ALS is a zero-temperature method, and so it is subject to replica symmetry breaking. AMP on the other hand is Bayes-optimal and thus operates at the Nishimori temperature [31]. At this temperature the system does not exhibit replica symmetry breaking, and the true global ground state can be found in the easy regime, when $\Delta < \Delta_{alg}$.

# 5    Summary

In summary, our work partially bridges the gap between theory and practice by creating new AMP algorithms that can flexibly assign different priors to different modes of a high-order tensor, thereby enabling AMP to handle arbitrarily shaped high order tensors that actually occur in the wild. Moreover, our theoretical analysis reveals interesting new phenomena governing how irregular tensor shapes can strongly affect inference performance and the positions of phase boundaries, and highlights the special, non-generic properties of cubic tensors. Finally, we hope the superior performance of our flexible AMP algorithms relative to ALS will promote the adoption of AMP in the wild. Code to reproduce all simulations presented in this paper is available at *https://github.com/ganguli-lab/tensorAMP*.

## Acknowledgments

We thank Alex Williams for useful discussions. We thank the Center for Theory of Deep Learning at the Hebrew University (J.K), and the Burroughs-Wellcome, McKnight, James S.McDonnell, and Simons Foundations, and the Office of Naval Research and the National Institutes of Health (S.G) for support.

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
