[Supplementary Material]

# Supplementary material: Statistical mechanics of low-rank tensor decomposition

**Jonathan Kadmon**
Department of Applied Physics, Stanford University
kadmonj@stanford.edu

**Surya Ganguli**
Department of Applied Physics, Stanford University and Google Brain, Mountain View, CA
sganguli@stanford.edu

In the following sections, we derive the approximate message passing (AMP) algorithms for arbitrary low-rank tensors. The derivation follows similar lines as the derivation in [3, 9] for the matrix $p = 2$ case, only here the model is generalized for higher modes p>2. We then derive dynamical mean field theory (also known as state evolution) and find the phase transition of the inference problem using non-linear analysis on the recursive dynamical equations of the order parameters. Lastly, we explicitly solve the equations for several simple examples of mode-3 tensors with a mixture of different prior distributions. For notational simplicity, we start the derivation assuming all modes are of the same dimension, $N$, which we assume to be in the thermodynamic limit, $N \to \infty$. Then, we will generalize for noncubic tensors. We emphasize that even in the cubic case the tensor is non-symmetric, and each mode is independent and is iid drawn from a prior distribution, which is potentially different for each mode.

## 1 Message passing on factorized graph

### 1.1 Factor graph for tensor decomposition – notations

We consider a given low-rank tensor

$$w_a^0 = \frac{1}{N^{\frac{p-1}{2}}} \sum_{\rho=1}^{r} \prod_{\alpha} x_{\alpha i}^{0\rho}. \tag{1}$$

Here the vectors $\mathbf{x}_{\alpha i}^0 \in \mathbb{R}^r$ denote the *ground-truth* values to the estimation problem. Each entry of the tensor is denoted with a lower-case latin letter $\{a, b, c, ...\}$. The notation stands for the set of $p$ indices that define that tensor element,

$$a = \{i_1, i_2, ... i_p\}. \tag{2}$$

However, we only have access to noisy measurements of the ground-truth vectors, denoted by

$$Y_a = w_a^0 + \sqrt{\Delta}\epsilon_a, \tag{3}$$

where $\epsilon_a$ is a random tesnsor, whose elements are i.i.d. gaussians with zero mean and unit variance. We assume no covariance between two measurements, $\mathbb{E}(\epsilon_a \epsilon_b) = 0 \; \forall a \neq b$.

The goal of the low rank decomposition is to find the estimators $\hat{\mathbf{x}}_{\alpha i} \in \mathbb{R}^r$ that minimize the mean square error

$$\hat{x} = \arg\min_x \sum_{\alpha} \sum_i \left( x_{\alpha i} - x_{\alpha i}^0 \right)^2.$$

To solve the Bayesian inference problem, using message passing, we frame it as a bipartite graphical model. Each of the variable nodes corresponds to an estimator $\mathbf{x}_{\alpha i}$ [See figure 1.b in the main text]. We use the notation $\partial a$ to denote all the neighboring nodes to $a$ and the notation $\partial a \backslash i\alpha$ to denote all the neighboring variable nodes adjacent to $a$, excluding the node $\alpha i$. The cardinality of the set of all factor points is $|\{a\}| = N^P$.

Each variable point on the graph is connected to $N^{p-1}$ factor nodes. The set of neighboring factor nodes that is connected to the variable node $\alpha i$ is denoted as

$$\partial \alpha i = \{a | \alpha i \in a\}. \tag{4}$$

## 1.2 Weakly connected graph

An underlying assumption in belief propagation and message passing algorithms is that the incoming messages into each node are statistically independent. It can be achieved, for example in sparse graphs, where at each node the graph can be approximately considered as a tree (directed acyclic graph), without recurring loops. In the physics literature such approximation is often referred to as Bethe Lattice. In the current model this is possible due to the scaling of individual elements, $w \sim N^{(1-p)/2}$, as defined in Eq. (1) in the main text. Since correlations in the messages are due to loops in the underlying graph, that pass through several nodes, we neglect them when the interactions are sufficiently weak [6].

## 1.3 Message passing

We start by defining two different types of messages (beliefs): one for messages outgoing from a variable node into a factor node $\eta$. Messages going from factor nodes into variable nodes are denoted by $\tilde{\eta}$. Messages are the marginal probabilities at each node, measuring the posterior probability density of the estimator at that source node. A message outgoing from the variable node $\alpha i$ to a factor node $a$ can be written in terms of the product of messages originating from all its connected nodes excluding $a$,

$$\eta_{\alpha i \to a}(\mathbf{x}_{\alpha i}) = \frac{P_\alpha(\mathbf{x}_{\alpha i})}{\mathcal{Z}_{\alpha i \to a}} \prod_{b \in \partial \alpha i \backslash a} \tilde{\eta}_{b \to \alpha i}(\mathbf{x}_{\alpha i}). \tag{5}$$

The denominator $\mathcal{Z}_{\alpha i \to a}$ is a normalization factor

$$\mathcal{Z}_{\alpha i \to a} = Tr_{x_{\alpha i}} P_\alpha(\mathbf{x}_{\alpha i}) \prod_{b \in \partial \alpha i \backslash a} \tilde{\eta}_{b \to \alpha i}(\mathbf{x}_{\alpha i}).$$

Incoming messages into variable nodes are obtained by marginalizing over distribution over all the messages. A message outgoing from a factor node into a variable node is given by

$$\tilde{\eta}_{b \to \alpha i}(\mathbf{x}_{\alpha i}) = \frac{1}{\mathcal{Z}_{b \to \alpha i}} \prod_{\beta j \in \partial b \backslash \alpha} Tr_{x_{\beta j}} \eta_{\beta j \to b}(\mathbf{x}_{\beta j}) \exp g\left(Y_b, N^{\frac{1-p}{2}} w_b\right), \tag{6}$$

where

$$w_b \equiv \sum_{\rho=1}^{r} \prod_\alpha x_{\alpha i}^\rho. \tag{7}$$

The normalization factor in the denominator of eq. (7) is given by

$$\mathcal{Z}_{b \to \alpha i} = Tr_{x_{\alpha i}} \prod_{\beta j \in \partial b \backslash \alpha} Tr_{x_{\beta j}} \eta_{\beta j \to b}(\mathbf{x}_{\beta j}) \exp g\left(Y_b, N^{\frac{1-p}{2}} w_b\right). \tag{8}$$

The cost function $g(\cdot)$ at the exponent can be expanded as a power series in $N$

$$\tilde{\eta}_{b \to \alpha i}(\mathbf{x}_{\alpha i}) = \frac{1}{\mathcal{Z}_{b \to \alpha i}} \prod_{\beta j \in \partial b \backslash \alpha} Tr_{x_{\beta j}} \eta_{\beta j \to b}(\mathbf{x}_{\beta j}) \times$$
$$\exp\left[g(Y_0, 0)\left(1 + \frac{1}{N^{(p-1)/2}} S_b w_b + \frac{1}{N^{p-1}}\left(R_b - S_b^2\right) w_b^2 + \mathcal{O}\left(\frac{1}{N^{3(p-1)/2}}\right)\right)\right] \tag{9}$$

where $S_b$ and $R_b$ are the first and second derivative of the cost function $g(Y, w)$ evaluated at $Y_b$ and $w_b = 0$:

$$S_b \equiv \left. \frac{\partial g(Y_b, w_b)}{\partial w} \right|_{w_b=0} \tag{10}$$

$$R_b \equiv \left( \left. \frac{\partial g(Y_b, w_b)}{\partial w} \right|_{w_b=0} \right)^2 + \left. \frac{\partial^2 g(Y_b, w_b)}{\partial w_b^2} \right|_{w_b=0} \tag{11}$$

**Belief propagation**

The mean values of outgoing messages from variable node $\alpha i$ into factor node $a$, are obtained by integrating over the marginal probabilities $\eta_{\alpha i \to a}$:

$$\hat{\mathbf{x}}_{\alpha i \to i} = \int dx_{\alpha i} \eta_{\alpha i \to a}(\mathbf{x}_{\alpha i}) \mathbf{x}_{\alpha i}^T \in \mathbb{R}^r. \tag{12}$$

Note that we have used the transpose of the vector $\mathbf{x}_{\alpha i}^T$, which will become useful for the notation below. Their covariance matrix is equal to

$$\hat{\sigma}_{\alpha i \to a} = \int d\mathbf{x}_{\alpha i} \eta_{\alpha i \to a}(\mathbf{x}_{\alpha i}) \mathbf{x}_{\alpha i} \mathbf{x}_{\alpha i}^T - \hat{\mathbf{x}}_{\alpha i \to i} \hat{\mathbf{x}}_{\alpha) \to i}^T \in \mathbb{R}^{r \times r}. \tag{13}$$

Using the first- and second-order statistics, we can write explicit expressions for the moments of $w_b$ appearing in the expansion (9) above. The first moment reads

$$\prod_{\beta j \in \partial b \backslash \alpha} \int d\mathbf{x}_{\beta i} \eta_{\beta) \to b}(\mathbf{x}_{\beta j}) w_b = \prod_{\beta j \in \partial b \backslash \alpha} \int dx_{\beta i} \eta_{\beta j \to b}(\mathbf{x}_{\beta j}) \sum_{\rho=1}^r \prod_{\beta j \in \partial b} x_{\beta j}^\rho$$

$$= \mathbf{x}_{\alpha i}^T \prod_{\beta j \in \partial b \backslash \alpha}^{\odot} \hat{\mathbf{x}}_{\beta j \to b}. \tag{14}$$

Similarly, the second moment, $w_b^2$, is given by

$$\prod_{\beta j \in \partial b \backslash \alpha} \int d\mathbf{x}_{\beta i} \eta_{\beta j \to b}(\mathbf{x}_{\beta j}) w_b^2 = \int \prod_{\beta j \in \partial b \backslash \alpha} d\mathbf{x}_{\beta i} \eta_{\beta j \to b}(\mathbf{x}_{\beta j}) \prod_{\beta j \in \partial b}^{\odot} x_{\beta j}^{\rho T} \prod_{\gamma k \in \partial b}^{\odot} x_{\gamma k}^\rho =$$

$$\mathbf{x}_{\alpha i}^T \prod_{\beta j \in \partial b \backslash \alpha}^{\odot} \left( \sigma_{\beta j \to b} + \hat{\mathbf{x}}_{\beta j \to b} \hat{\mathbf{x}}_{\beta j \to b}^T \right) \mathbf{x}_{\alpha i}. \tag{15}$$

Introducing the explicit moments back into eq. (9), the incoming messages into variable nodes are given by

$$\tilde{\eta}_{b \to \alpha i}(x_{\alpha i}) = \frac{e^{g(Y_b, 0)}}{\mathcal{Z}_{b \to \alpha i}} \left[ 1 + \frac{1}{N^{(p-1)/2}} S_b \mathbf{x}_{\alpha i}^T \prod_{\beta j \in \partial b \backslash \alpha}^{\odot} \hat{\mathbf{x}}_{\beta j \to b} + \right.$$

$$\left. \frac{1}{N^{p-1}} \left( R_b - S_b^2 \right) x_{\alpha i}^T \prod_{\beta j \in \partial b \backslash \alpha}^{\odot} \left( \sigma_{\beta j \to b} + \mathbf{x}_{\beta j \to b} \mathbf{x}_{\beta j \to b}^T \right) \mathbf{x}_{\alpha i} \right] + O\left( \frac{1}{N^{3(p-1)/2}} \right). \tag{16}$$

Since we are interested in the marginals in the variable nodes, we can replace this result in the expression for messages outgoing from a variable node (5), we obtain

$$\eta_{\alpha i \to a}(\mathbf{x}_{\alpha i}) = \frac{P_\alpha(\mathbf{x}_{\alpha i})}{\mathcal{Z}_{\alpha i \to a}} \prod_{b \in \partial \alpha i \backslash a} \tilde{\eta}_{b \to \alpha i}(\mathbf{x}_{\alpha i}) = \frac{P_\alpha(\mathbf{x}_{\alpha i})}{\mathcal{Z}_{\alpha i \to a}} \frac{e^{N g(Y_b, 0)}}{\prod_{b \in \partial \alpha i \backslash a} \mathcal{Z}_{b \to \alpha i}} \times$$

$$\exp \sum_{b \in \partial \alpha i \backslash a} \left[ \frac{1}{N^{p-1}} \left( R_b - S_b^2 \right) x_{\alpha i}^T \prod_{\beta j \in \partial b \backslash \alpha}^{\odot} \left( \sigma_{\beta j \to b} + \mathbf{x}_{\beta j \to b} \mathbf{x}_{\beta j \to b}^T \right) \mathbf{x}_{\alpha i} \right]. \tag{17}$$

Note that $g(Y_b, 0)$ is a constant and can be absorbed into the normalization function. We define the two order parameters

$$\mathbf{u}_{\alpha i \to a}^T = \frac{1}{N^{(p-1)/2}} \sum_{b \in \partial \alpha i \backslash a} S_b \prod_{\beta j \in \partial b \backslash \alpha i}^{\odot} \hat{\mathbf{x}}_{\beta j \to b}^T \in \mathbb{R}^r, \tag{18}$$

and

$$A_{\beta j \to b} = \frac{1}{N^{p-1}} \sum_{b \in \partial \alpha i \backslash a} \left[ \prod_{\beta j \in \partial b \backslash \alpha i}^{\odot} S_b^2 \mathbf{x}_{\beta j \to b} \mathbf{x}_{\beta j \to b}^T \right.$$

$$\left. - R_b \prod_{\beta j \in \partial b \backslash \alpha i}^{\odot} \left( \sigma_{\beta j \to b} + \mathbf{x}_{\beta j \to b} \mathbf{x}_{\beta j \to b}^T \right) \right] \in \mathbb{R}^{r \times r}. \tag{19}$$

Using the order parameters we rewrite equation (17) as

$$\eta_{\alpha i \to a}(\mathbf{x}_{\alpha i}) = \frac{P_\alpha(\mathbf{x}_{\alpha i})}{\mathcal{Z}_{\alpha i \to a}} \prod_{b \in \partial \alpha i \backslash a} \exp \left( -\mathbf{x}_{\alpha i}^T A_{\beta j \to b} \mathbf{x}_{\alpha i} + \mathbf{u}_{\beta j \to b}^T \mathbf{x}_{\alpha i} \right). \tag{20}$$

The normalization, or partition function $\mathcal{Z}_{\alpha i \to a}$, can be written in terms of the order parameters $\mathbf{u}_{\alpha i \to a}^T$ and $A_{\beta j \to b}$ as

$$\mathcal{Z}_{\alpha i \to a} = Tr_{x_{\alpha i}} P_\alpha(\mathbf{x}_{\alpha i}) \prod_{b \in \partial \alpha i \backslash a} \exp \left( -\mathbf{x}_{\alpha i}^T A_{\beta j \to b} \mathbf{x}_{\alpha i} + \mathbf{u}_{\beta j \to b}^T \mathbf{x}_{\alpha i} \right). \tag{21}$$

Finally, the moments of the local variables $x_{\alpha i}$ with distribution $\eta_{\alpha i \to a}(\mathbf{x}_{\alpha i})$ can be found directly from the partition functions $\mathcal{Z}_{\alpha i \to a}$ by standard derivations. The mean is given by

$$\hat{x}_{\alpha i \to a} = \frac{\partial}{\partial \mathbf{u}_{\alpha i \to a}} \mathcal{Z}_{\alpha i \to a}(A_{\alpha i \to a}, \mathbf{u}_{\alpha i \to a}) \equiv f(A_{\alpha i \to a}, \mathbf{u}_{\alpha i \to a}), \tag{22}$$

and the covariance matrices are

$$\sigma_{\alpha i \to a} = \frac{\partial^2}{\partial \mathbf{u}_{\alpha i \to a} \partial \mathbf{u}_{\alpha i \to a}^T} \mathcal{Z}_{\alpha i \to a}(A_{\alpha i \to a}, \mathbf{u}_{\alpha i \to a})$$

$$= \frac{\partial}{\partial \mathbf{u}_{\alpha i \to a}} f(A_{\alpha i \to a}, \mathbf{u}_{\alpha i \to a}). \tag{23}$$

## 1.4 AMP algorithms

The mean-field equations, describing the equilibrium of the local estimators can be used to iteratively into an algorithm by iteratively calculating the statistics of the messages given their estimators using eq. (18) and (19) and then reevaluating the estimators $\hat{x}$ and $\sigma$ using eq. (22) and (23). Defining the upper-script $t$ denoting the time step of the algorithm iteration, we can write the iterative equations as

$$\mathbf{u}_{\alpha i \to a}^t = \frac{1}{N^{(p-1)/2}} \sum_{b \in \partial \alpha i \backslash a} S_b \prod_{\beta j \in \partial b \backslash \alpha i}^{\odot} \hat{x}_{\beta j \to b}^t \tag{24}$$

$$A_{\alpha i \to a}^t = \frac{1}{N^{p-1}} \sum_{b \in \partial \alpha i \backslash a} \left[ S_b^2 \prod_{\beta j \in \partial b \backslash \alpha i}^{\odot} \hat{\mathbf{x}}_{\beta j \to b}^t \hat{\mathbf{x}}_{\beta j \to b}^{tT} \right. \tag{25}$$

$$\left. - R_b \prod_{\beta j \in \partial b \backslash \alpha i}^{\odot} \left( \sigma_{\beta j \to b}^t + \hat{\mathbf{x}}_{\beta j \to b}^t \hat{\mathbf{x}}_{\beta j \to b}^{tT} \right) \right] \tag{26}$$

$$\hat{x}_{\alpha i \to a}^{t+1} = \frac{\partial}{\partial \mathbf{u}_{\alpha i \to a}^t} \log \mathcal{Z}_{\alpha i \to a}(A_{\alpha i \to a}^t, \mathbf{u}_{\alpha i \to a}^t) \tag{27}$$

$$\sigma_{\alpha i \to a}^{t+1} = \frac{\partial^2}{\partial \mathbf{u}_{\alpha i \to a}^t \partial \mathbf{u}_{\alpha i \to a}^{tT}} \log \mathcal{Z}_{\alpha i \to a}(A_{\alpha i \to a}^t, \mathbf{u}_{\alpha i \to a}^t) \tag{28}$$

## 1.5 Approximate message passing – local mean-field approximation for the messages

In the equations above (24)-(28), the number of overall messages (and thus calculations) scale with the number of edges in the factorized graph, i.e. as $\mathcal{O}(N^P)$. However, the dependence of each message on the state of the target node is weak. Therefore, the values of $\mathbf{u}_{\alpha i \to a}^t$ and $A_{\alpha i \to a}^t$ are very close to their mean, when marginalized over all target nodes $a$. The local deviations about that mean scale as $N^{(1-p)/2}$. For that reason, we can consider the statistics of all outgoing messages from each node (i.e., average over all the adjacent edges), and assume small fluctuations due to the state of the targets. This procedure is essentially performing mean-field approximation at every node. The result will be the AMP equations which scale with the number of variable nodes $PN$, rather than with the number of edges in the graph. In physics, this analogous to the cavity method.

To apply this reasoning to the equations, we define the order parameters $A_{\alpha i}$ and $\mathbf{u}_{\alpha i}$, which explicitly exclude the dependence of the target node:

$$\mathbf{u}_{\alpha i}^t = \frac{1}{N^{(p-1)/2}} \sum_{b \in \partial \alpha i} S_b \prod_{\beta j \in \partial b}^{\odot} \hat{\mathbf{x}}_{\beta j \to b}^t, \tag{29}$$

$$A_{\alpha i}^t = \frac{1}{N^{p-1}} \sum_{b \in \partial \alpha i} \left[ S_b^2 \prod_{\beta j \in \partial b \setminus \alpha i}^{\odot} \hat{\mathbf{x}}_{\beta j \to b}^t \hat{\mathbf{x}}_{\beta j \to b}^{tT} - R_b \prod_{\beta j \in \partial b \setminus \alpha i}^{\odot} \left( \sigma_{\beta j \to b}^t + \hat{\mathbf{x}}_{\beta j \to b}^t \hat{\mathbf{x}}_{\beta j \to b}^{tT} \right) \right]. \tag{30}$$

The difference between the non-directed and the directed messages is the component that depends on the target node $S_a$. For the mean-messages, the correction terms scale as $O(N^{(1-p)/2})$, and is given by

$$\delta \mathbf{u}_{\alpha i \to a}^t = \mathbf{u}_{\alpha i}^t - \mathbf{u}_{\alpha i \to a}^t = \frac{1}{N^{(p-1)/2}} S_a \prod_{\beta j \in \partial a \setminus \alpha i}^{\odot} \hat{\mathbf{x}}_{\beta j \to a}^t. \tag{31}$$

For the fluctuations in the local messages about their mean, the correction term scales as

$$A_{\alpha i}^t - A_{\alpha i \to a}^t \sim \mathcal{O}(N^{1-p}), \tag{32}$$

and we will be neglecting it.

To transform the equations for the local messages statistics, to use only the *target-agnostic* variables, $\hat{x}_{\beta j}^t$ and $\sigma_{\beta j}^t$, we calculate the difference between the two mean values

$$\delta \hat{x}_{\alpha i \to a}^t = \hat{\mathbf{x}}_{\alpha i}^t - \hat{\mathbf{x}}_{(\alpha i) \to a}^t = f\left( A_{\alpha i \to a}^{t-1}, \mathbf{u}_{\alpha i \to a}^{t-1} \right) - f\left( A_{\alpha i}^{t-1}, \mathbf{u}_{\alpha i}^{t-1} \right). \tag{33}$$

We develop the second term on the RHS to linear order in the small parameter of the difference $\delta \mathbf{u}_{\alpha i \to a}$, and note that the leading order cancel with the first term in the RHS above, yielding

$$\delta \hat{x}_{\alpha i \to a}^t = f\left( A_{\alpha i}^{t-1}, \mathbf{u}_{\alpha i}^{t-1} \right) + \frac{\partial}{\partial \mathbf{u}} f\left( A_{\alpha i}^{t-1}, \mathbf{u}_{\alpha i}^{t-1} \right) \left( \mathbf{u}_{\alpha i \to a}^{t-1} - \mathbf{u}_{\alpha i}^{t-1} \right) - f\left( A_{\alpha i}^{t-1}, \mathbf{u}_{\alpha i}^{t-1} \right) =$$

$$\sigma_{\alpha i}^t \left( \mathbf{u}_{\alpha i \to a}^{t-1} - \mathbf{u}_{\alpha i}^{t-1} \right) = \sigma_{\alpha i}^t \frac{1}{N^{(p-1)/2}} S_a \prod_{\beta j \in \partial a \setminus \alpha i}^{\odot} \hat{\mathbf{x}}_{\beta, j}^{t-1}. \tag{34}$$

Using eq. (31) and (34), we can write an expression for the node-average local messages,

$$\mathbf{u}_{\alpha i}^t = \frac{1}{N^{(p-1)/2}} \sum_{b \in \partial \alpha i} S_b \prod_{\beta j \in \partial b \setminus \alpha i}^{\odot} \left( \hat{\mathbf{x}}_{\beta j}^t - \delta \hat{\mathbf{x}}_{\beta j \to b}^t \right). \tag{35}$$

Expanding the product of the $\beta j$ factors, and keeping terms up to linear order in the small difference $\delta x$, we obtain

$$\mathbf{u}_{\alpha i}^t = \frac{1}{N^{(p-1)/2}} \sum_{b \in \partial \alpha i} S_b \left[ \prod_{\beta j \in \partial b \setminus \alpha i}^{\odot} \hat{\mathbf{x}}_{\beta j}^t - \sum_{\beta j \in \partial b \setminus \alpha i} \delta \hat{\mathbf{x}}_{\beta j \to b}^t \prod_{\gamma k \in \partial b \setminus \alpha i, \beta j}^{\odot} \hat{\mathbf{x}}_{\gamma k}^t \right] + O\left( \frac{1}{N^{(p-1)}} \right). \tag{36}$$

The first correction for the above, involves the quadratic terms in the expansion of (35). The mixed terms involve the values at time $t$ **and** at **t**ime $t-1$, which originate from the expansion of $\delta x$ in (34). The mixed term is given by

$$\frac{1}{N^{(p-1)}} \sum_{b \in \partial \alpha i} S_b^2 \sum_{\beta j \in \partial b \backslash \alpha i} \sigma_{\beta j}^t \prod_{\gamma,k \in \partial b \backslash \beta j}^{\odot} \hat{\mathbf{x}}_{\gamma k}^{t-1} \prod_{\gamma k \in \partial b \backslash \alpha i, \beta j}^{\odot} \hat{\mathbf{x}}_{\gamma k}^t =$$
$$\frac{1}{N^{(p-1)}} \hat{x}_{\alpha i}^{t-1} \sum_{b \in \partial \alpha i} S_b^2 \sum_{\beta j \in \partial b \backslash \alpha i} \sigma_{\beta j}^t \prod_{\gamma k \in \partial b \backslash \alpha i, \beta j}^{\odot} \hat{\mathbf{x}}_{\gamma k}^t \hat{\mathbf{x}}_{\gamma k}^{t-1}. \quad (37)$$

This expression, which couples the dynamical variable $\hat{x}$ into its previous time step is an *Onsager response term*. It reflects the changes to the fields of the nodes surrounding the node $\alpha i$ due to the activity of the node $\alpha i$ in the previous time step.

Importantly, up until this point, we have not yet used the assumption of Bayes optimality, nor have we used the Nishimori identities that follow the Bayes-optimal assumption. Consequently, algorithms based on the approximate message-passing above should be general and do not require the Bayes-optimal assumption. In the following section, we consider simplification due to the Bayes-optimal assumption. Beyond simplification of the mathematical expressions, it will allow us to systematically derive a dynamical mean-field theory for the errors in section 2.

## 1.6 Simplifications for Bayes-optimal settings

The covariance matrix in the Bayes-optimal case can be much simplified. First, one can show, using the Nishimori identities at the equilibrium, that

$$\langle R_b \rangle \equiv \left\langle \left. \frac{\partial g(Y_b, w)}{\partial w} \right|_{w=0} \right|^2 \right\rangle + \left\langle \left. \frac{\partial^2 g(Y_b, w)}{\partial w^2} \right|_{w=0} \right\rangle = 0. \quad (38)$$

Here the angular brackets denote averaging over the posterior. The posterior variance of $S_b$ is given by the Fisher information of the output channel $\mathbb{E}_{post}\left[S_b^2\right] = \frac{1}{\Delta}$. Furthermore, in Bayes-optimal setting all samples from the equilibrium ensemble are similar, the sum over nodes becomes self-averaging and so

$$\sum_b S_b^2 = \mathbb{E}_{Post}\left[S_b^2\right] = \frac{1}{\Delta}. \quad (39)$$

Using the above simplifications, the covariance matrix of the messages can be written as

$$A_{\alpha i}^t = \frac{1}{N^{p-1}} \frac{1}{\Delta} \sum_{b \in \partial \alpha i} \prod_{\beta j \in \partial b \backslash \alpha i}^{\odot} \hat{\mathbf{x}}_{\beta j \to b}^t \hat{\mathbf{x}}_{\beta j \to b}^{tT} = \frac{1}{\Delta} \prod_{\beta \neq \alpha}^{\odot} \left( \frac{1}{N} \sum_{j=1}^N \hat{\mathbf{x}}_{\beta j}^t \hat{\mathbf{x}}_{\beta j}^{tT} \right) \equiv A_\alpha^t. \quad (40)$$

Importantly, the covariance matrix does not depend on the specific node $i$, but it does depend on the mode of the tensor $\alpha$. This is a significant difference from the algorithms for symmetric-tensor decomposition, for which $A^t$ was uniform for *all* nodes in the factor graph [4].

The Onsager term, which appears in the iterative mean-field equations for the local mean of the messages can also be simplified under the Bayes-optimal setting. Using some algebra, the Onsager correction becomes

$$\frac{1}{N^{(p-1)}} \hat{\mathbf{x}}_{\alpha i}^{t-1} \sum_{b \in \partial \alpha i} S_b^2 \sum_{\beta j \in \partial b \backslash \alpha i} \sigma_{\beta j}^t \prod_{(\gamma,k) \in \partial b \backslash \alpha i, (\beta j)}^{\odot} \hat{\mathbf{x}}_{\gamma k}^t \hat{\mathbf{x}}_{\gamma k}^{t-1} =$$
$$\frac{1}{\Delta N} \hat{\mathbf{x}}_{\alpha i}^{t-1} \sum_{b \in \partial \alpha i} \sum_{\beta j \in \partial b \backslash \alpha i} \sigma_{\beta j}^t \prod_{\gamma \neq \alpha, \beta}^{\odot} \left( \frac{1}{N} \sum_k \hat{\mathbf{x}}_{\gamma k}^t \hat{\mathbf{x}}_{\gamma k}^{t-1} \right) =$$
$$\frac{1}{\Delta N} \hat{\mathbf{x}}_{\alpha i}^{t-1} \sum_{\beta \neq \alpha} \sum_j \sigma_{\beta j}^t \odot \prod_{\gamma \neq \alpha, \beta}^{\odot} \left( \frac{1}{N} \sum_k \hat{\mathbf{x}}_{\gamma k}^t \hat{\mathbf{x}}_{\gamma k}^{t-1} \right) \equiv$$
$$\frac{1}{\Delta N} \hat{\mathbf{x}}_{\alpha i}^{t-1} \sum_{\beta \neq \alpha} \sum_j \sigma_{\beta j}^t \odot D_{\alpha \beta}^t, \quad (41)$$

where

$$D_{\alpha\beta}^t = \prod_{\gamma \neq \alpha, \beta}^{\odot} \left( \frac{1}{N} \sum_k \hat{\mathbf{x}}_{\gamma k}^t \hat{\mathbf{x}}_{\gamma k}^{t-1} \right). \tag{42}$$

Finally, we write the simplified AMP equations as

$$\mathbf{u}_{\alpha i}^t = \frac{1}{N^{(p-1)/2}} \sum_{b \in \partial \alpha i} S_b \prod_{\beta j \in \partial b \setminus \alpha i}^{\odot} \hat{\mathbf{x}}_{\beta j}^t - \frac{1}{\Delta} \hat{\mathbf{x}}_{\alpha i}^{t-1} \sum_{\beta \neq \alpha} \Sigma_{\beta}^t \odot D_{\alpha\beta}^t \tag{43}$$

$$A_\alpha^t = \frac{1}{\Delta} \prod_{\beta \neq \alpha}^{\odot} \left( \frac{1}{N} \sum_{j=1}^N \hat{\mathbf{x}}_{\beta j}^t \hat{\mathbf{x}}_{\beta j}^{tT} \right) \tag{44}$$

$$\hat{\mathbf{x}}_{\alpha i}^{t+1} = \frac{\partial}{\partial \mathbf{u}_{\alpha i}^t} \log \mathcal{Z}_\alpha(A_\alpha^t, \mathbf{u}_{\alpha i}^t) \tag{45}$$

$$\sigma_{\alpha i}^{t+1} = \frac{\partial^2}{\partial \mathbf{u}_{\alpha i}^t \partial \mathbf{u}_{\alpha i}^{tT}} \log \mathcal{Z}_\alpha(A_\alpha^t, \mathbf{u}_{\alpha i}^t), \tag{46}$$

where the Onsager term is given by

$$D_{\alpha\beta}^t = \prod_{\gamma \neq \alpha, \beta}^{\odot} \left( \frac{1}{N} \sum_k \hat{\mathbf{x}}_{\gamma k}^t \hat{\mathbf{x}}_{\gamma k}^{t-1} \right) \tag{47}$$

$$\Sigma_\alpha^t = N^{-1} \sum_i \sigma_{\alpha i}^t, \tag{48}$$

and the partition function reads

$$\mathcal{Z}_\alpha(A_\alpha^t, \mathbf{u}_{\alpha i}^t) = \int d\mathbf{x} P_\alpha(\mathbf{x}) \exp \left[ \left( \mathbf{u}_{\alpha i}^T x - \mathbf{x}^T A_\alpha^t \mathbf{x} \right) \right]. \tag{49}$$

There are two parameters in these equations (apart from the prior distributions $P_\alpha(x)$. One is the Fisher information of the output channel, which is a global parameter that can tune the global dynamics. The other $S_b$, which is the Fisher score of the entry at $Y_b$. The last one is what yields the structure in the solution of the estimators.

## 2 Dynamic mean field theory (state evolution)

In the previous section we have derived the AMP algorithm for general tensors, and show their simplified form in the case of the Bayes-optimal assumption, where the priors are known, and the system follow Nishimori identities at equilibrium. These algorithms follow the iterative evolution of the estimators in each of the variable nodes in the factor graph. In order to analytically study the performance of the algorithm, we want to know how the mean error reduces from one iteration of the algorithm to the next. To do that, we derive a dynamical mean-field theory (also known as *state-evolution*). As mentioned above, following the Bayes-optimal assumption, the estimators are self-averaging; thus a mean-field description of the error is a good measure for the typical evolution of any given system.

We define an order parameter that measures the overlap between each of the underlying vectors of estimators $\hat{x}_\alpha^t \in \mathbb{R}^{p \times r}$ and the ground truth values $x_\alpha^0 \in \mathbb{R}^{p \times r}$. The *overlap matrix* is defined as

$$M_\alpha^t = \frac{1}{N} \sum_i^N \hat{\mathbf{x}}_{\alpha i}^t \mathbf{x}_{\alpha i}^{0T} \in \mathbb{R}^{r \times r}. \tag{50}$$

In total, there are $p$ matrices of dimensions $r \times r$, each for each mode of the tensor. In the Bayes-optimal regime, the ground-truth values can be replaced with any typical sample from the posterior distribution. Thus, in Bayes-optimal inference, $M_\alpha^t$ is also the typical covariance matrix of the estimators

$$\frac{1}{N} \sum_i^N \hat{\mathbf{x}}_{\alpha i}^t \hat{\mathbf{x}}_{\alpha i}^t = M_\alpha^t. \tag{51}$$

It follows that under the Bayes-optimality condition, $M_\alpha^t$ is a symmetric matrix.

To study the typical dynamics of the algorithm using the mean overlap, we derive yet another mean-field approximation, now on the spatial degrees of freedom – i.e., the nodes. Given the self-averaging property of the nodes under the Bayes-optimal setting, and using the central-limit theorem, we need to find the first two moments of the distribution of the local values $\mathbf{u}_{i\alpha}$ (note that $A_\alpha$ are already node-independent). Following the usual procedure of mean-field theory, we then close the equations self-consistently using the overlap parameter $M_\alpha^t$.

Using the definition of $\mathbf{u}_{\alpha i}$ from eq. (29), we average over the posterior $P_{out}$:

$$\mathbb{E}\left[\mathbf{u}_{\alpha i}^t\right] = \frac{1}{N^{(p-1)/2}} \sum_{b\in\partial\alpha i} \mathbb{E}\left[S_b \prod_{\beta j\in\partial b\backslash\alpha i}^{\odot} \hat{\mathbf{x}}_{\beta j\to b}^t\right] =$$

$$\frac{1}{N^{(p-1)/2}} \sum_{b\in\partial\alpha i} \int dY_b P_{out}(Y_b, w_b) \left.\frac{\partial \log P_{out}(Y_b \,|\, w)}{\partial w}\right|_{w=0} \prod_{\beta j\in\partial b\backslash\alpha i}^{\odot} \hat{\mathbf{x}}_{\beta j\to b}^t \tag{52}$$

Similar to the the approximation carried above for the AMP algorithms, we develop the posterior probability about $w = 0$, and keep only the leading terms,

$$P_{out}(Y_b, w_b) = P_{out}(Y_b, 0) + P_{out}(Y_b, 0)w_b \left(\frac{\partial \log P_{out}(Y_b, w)}{\partial w}\right)_{w=0} + O(w^2). \tag{53}$$

Carrying the integration in eq. (52), the leading order will vanish

$$\int dY_b P_{out}(Y_b, 0) \left.\frac{\partial \log P_{out}(Y_b \,|\, w)}{\partial w}\right|_{w=0} = 0, \tag{54}$$

which is the consequence of the Nishimori identities. In other words, in a Bayes-optimal setting, and when the interactions are weak, then the average value of the messages when averaged over the entire graph vanish to leading order. Intuitively, since the underlying graph is isotropic, we expect that the dynamics will be similar at every node on average.

Performing the integration on the next, quadratic, term in (52) we get

$$\mathbb{E}\left[\mathbf{u}_{\alpha i}^t\right] = \frac{1}{N^{(p-1)/2}} \times$$

$$\sum_{b\in\partial\alpha i} P_{out}(Y_b, 0)w_b \left(\frac{\partial \log P_{out}(Y_b, w)}{\partial w}\right)_{w=0}^2 \prod_{\beta j\in\partial b\backslash\alpha i}^{\odot} \hat{\mathbf{x}}_{\beta j\to b}^t =$$

$$\frac{1}{\Delta N^{(p-1)/2}} \sum_{b\in\partial\alpha i} w_b \prod_{\beta j\in\partial b\backslash\alpha i}^{\odot} \hat{\mathbf{x}}_{\beta j\to b}^t. \tag{55}$$

Note that the original tensor components, denoted by $w_a$ are the ground-truth in the context of the inference problem, and we can write

$$w_a = \frac{1}{N^{\frac{p-1}{2}}} \prod_{(\beta j)\in\partial a}^{\odot} \mathbf{x}_{\beta j}^0. \tag{56}$$

Replacing this into the expression for the expectation above we get

$$\mathbb{E}\left[\mathbf{u}_{\alpha i}^t\right] = \frac{1}{\Delta N^{(p-1)}} \sum_{b \in \partial \alpha i} \sum_{\rho=1}^{r} \prod_{(\beta j) \in \partial b}^{\odot} \mathbf{x}_{\beta j}^{0,\rho} \prod_{\beta j \in \partial b \backslash \alpha i}^{\odot} \hat{\mathbf{x}}_{\beta j}^{tT}$$

$$= \frac{1}{\Delta N^{(p-1)}} \sum_{b \in \partial \alpha i} \left(\mathbf{x}_{\alpha i}^0\right)^T \left(\prod_{\beta j \in \partial b \backslash \alpha i}^{\odot} \mathbf{x}_{\beta j}^0\right) \left(\prod_{\beta j \in \partial b \backslash \alpha i}^{\odot} \hat{\mathbf{x}}_{\beta j}^t\right)^T$$

$$= \frac{1}{\Delta} \left(\mathbf{x}_{\alpha i}^0\right)^T \prod_{\beta \neq \alpha}^{\odot} \left(\frac{1}{N} \sum_j^N \mathbf{x}_{\beta j}^0 \hat{\mathbf{x}}_{\beta j}^{tT}\right) = \frac{1}{\Delta} \left(\mathbf{x}_{\alpha i}^0\right)^T \prod_{\beta \neq \alpha}^{\odot} M_\beta^t. \quad (57)$$

Finally we can write

$$\mathbb{E}\left[\mathbf{u}_{\alpha i}^t\right] = \frac{1}{\Delta} \prod_{\beta \neq \alpha}^{\odot} M_\beta^t \mathbf{x}_{\alpha i}^0. \quad (58)$$

In a similar manner, we can calculate the covariance matrix of the mean-messages

$$cov\left[\mathbf{u}_{i\alpha}^t\right] = \sum_{a \in \partial \alpha i} \int dY_a P_{out}(Y_a, w) \left(\frac{\partial g(Y_a, w)}{\partial w}\right)_{w=0}^2 \frac{1}{N^{p-1}} \prod_{\beta j \in \partial a \backslash \alpha i}^{\odot} \hat{\mathbf{x}}_{\beta j}^t \hat{\mathbf{x}}_{\beta j}^{tT}. \quad (59)$$

Keeping the leading order after the expansion of the distribution $P_{out}$ for small $w$ we get

$$cov\left[\mathbf{u}_{i\alpha}^t\right] = \frac{1}{N^{p-1}\Delta} \sum_{a \in \partial \alpha i} \prod_{\beta j \in \partial a \backslash \alpha i}^{\odot} \hat{\mathbf{x}}_{\beta j}^t \hat{\mathbf{x}}_{\beta j}^{tT}. \quad (60)$$

In the Bayes-optimal setting this is equal to

$$cov\left[\mathbf{u}_{i\alpha}^t\right] = \frac{1}{\Delta} \prod_{\beta \neq \alpha}^{\odot} M_\beta^t. \quad (61)$$

While the mean-message $\mathbf{u}_{\alpha i}$ varies from node to node, the mean covariance (not to be confused with the *covariance of the mean* calculated above), $A_\alpha^t$, is node-independent, as we have established in the previous section. In the Bayes-optimal setting, where the Nishimori identities hold, it is equal to

$$A_\alpha^t = \frac{1}{\Delta} \prod_{\beta \neq \alpha}^{\odot} \left(\frac{1}{N} \sum_{j=1}^N \hat{\mathbf{x}}_{\beta j}^t \hat{\mathbf{x}}_{\beta j}^{tT}\right) = \frac{1}{\Delta} \prod_{\beta \neq \alpha}^{\odot} M_\beta^t. \quad (62)$$

Using the definition of the mean overlap in eq. (50), and eq. (45), we write

$$M_\alpha^t = \frac{1}{N} \sum_i^N \hat{\mathbf{x}}_{\alpha i}^t \mathbf{x}_{\alpha i}^{0T} = \frac{1}{N} \sum_i f_\alpha \left(A_\alpha^{t-1}, \mathbf{u}_{\alpha i}^{t-1}\right) \mathbf{x}_{\alpha i}^{0T}, \quad (63)$$

where

$$f_\alpha \equiv \frac{\partial}{\partial \mathbf{u}} \log \mathcal{Z}(A_\alpha, \mathbf{u}_\alpha), \quad (64)$$

and with the partition function

$$\mathcal{Z}_\alpha(A_\alpha, \mathbf{u}_\alpha) = \int d\mathbf{x} P_\alpha(\mathbf{x}) \exp\left[\left(\mathbf{u}_\alpha^T \mathbf{x} - x^T A_\alpha^t \mathbf{x}\right)\right]. \quad (65)$$

Replacing the average over all nodes $i$ in (63) with the expectation, we write an iterative update equation for the order parameter $M_\alpha^t$,

$$M_\alpha^{t+1} = \int d\boldsymbol{x}_\alpha^0 P_\alpha(\boldsymbol{x}_\alpha^0) \mathbb{E}_z \left[ f_\alpha \left( \frac{1}{\Delta} \prod_{\beta \neq \alpha}^{\odot} M_\beta^t, \frac{1}{\Delta} \prod_{\beta \neq \alpha}^{\odot} M_\beta^t x_{\alpha i}^0 + \frac{1}{\sqrt{\Delta}} \left( \prod_{\beta \neq \alpha}^{\odot} M_\beta^t \right)^{\frac{1}{2}} z \right) \mathbf{x}_{\alpha i}^{0T} \right].$$
$$(66)$$

Here, $z \in \mathbb{R}^r$ are random variables with standard normal distribution. The expectation in the RHS of eq. (66) is over two random variables: First are expected values for the underlying ground-truth $\boldsymbol{x}_\alpha^0$, which follows the prior distribution $P_\alpha$; The second is of a standard gaussian variable $z$, which represent the node-to-node fluctuations in the local mean-messages, with mean $\frac{1}{\Delta} \prod_{\beta \neq \alpha}^{\odot} M_\beta^t x_{\alpha i}^0$ and covariance matrix $\frac{1}{\Delta} \prod_{\beta \neq \alpha}^{\odot} M_\beta^t$.

The final overlap values of the iterative algorithms are given by the stable fixed points of the dynamic equations defined in (66). These can be obtained by finding the solutions $M_\alpha^*$ for the $p$ equations

$$
M_\alpha^* = \int d\boldsymbol{x}_\alpha^0 P_\alpha(\boldsymbol{x}_\alpha^0) \mathbb{E}_z \left[ f_\alpha \left( \frac{1}{\Delta} \prod_{\beta \neq \alpha}^{\odot} M_\beta^*, \frac{1}{\Delta} \prod_{\beta \neq \alpha}^{\odot} M_\beta^* x_{\alpha i}^0 + \frac{1}{\sqrt{\Delta}} \left( \prod_{\beta \neq \alpha}^{\odot} M_\beta^* \right)^{\frac{1}{2}} z \right) \mathbf{x}_{\alpha i}^{0T} \right].
$$
(67)

**Mean square error**    The real quantity of interest is the mean square error (MSE) of the estimate. This can be easily obtained from the mean overlap at any time-step of the algorithm using.

$$
MSE_\alpha^t = \frac{1}{\sigma_\alpha^2} Tr \left[ \mathbb{E}_{P_\alpha} \left[ \mathbf{x^0}_\alpha \mathbf{x}_\alpha^{0T} \right] - M_\alpha^t \right].
$$
(68)

Finally, the expected error of the AMP algorithms, once it has converged is given by

$$
MSE_\alpha^{AMP} = \frac{1}{\sigma_\alpha^2} Tr \left[ \mathbb{E}_{P_\alpha} \left[ \mathbf{x^0}_\alpha \mathbf{x}_\alpha^{0T} \right] - M_\alpha^* \right].
$$
(69)

## 3    Convergence of the AMP algorithms

Approximate message passing, and belief-propagation algorithms in general are known to have convergence issues (see for example [7, 1, 2, 8, 5]). A typical naive implementation of the algorithms will reduce the overall mean square error of the estimator, $MSE^t$. However, at some point, $MSE^t$ will start increasing and may diverge to large deviations from the ground-truth values or will oscillate about some fixed value. Loosely speaking, the step size of the iterative update equations (45) and (46) is too big, and the algorithm may 'overshoot' the MMSE estimator. One possible way to correct this behavior (see e.g., [8] and reference therein) is to reduce the step size. Since the differential change to $x^t$ and $\sigma^t$ is proportional to derivatives of the partition function in (45) and (46), a good normalization scheme could use an energy estimation of the configuration at time-step $t$. To do this, one can evaluate the Bethe free energy at every time step [8, 5, 3]. However, since this report does not focus on possible implementations of the algorithms, it is sufficient to use a simpler – and potentially less efficient – scheme, using fixed step-size reduction, or *damping*.

To implement the fixed damping algorithm, eq (45) and (46) can be rewritten as

$$
\hat{\mathbf{x}}_{\alpha i}^{t+1} = \lambda \hat{\mathbf{x}}_{\alpha i}^{t+1} + (1 - \lambda) \frac{\partial}{\partial \mathbf{u}_{\alpha i}^t} \log \mathcal{Z}_\alpha(A_\alpha^t, \mathbf{u}_{\alpha i}^t)
$$
(70)

$$
\sigma_{\alpha i}^{t+1} = \lambda \sigma_{\alpha i}^{t+1} + (1 - \lambda) \frac{\partial^2}{\partial \mathbf{u}_{\alpha i}^t \partial \mathbf{u}_{\alpha i}^{tT}} \log \mathcal{Z}_\alpha(A_\alpha^t, \mathbf{u}_{\alpha i}^t)
$$
(71)

where $0 \leq \lambda < 1$ is the damping coefficient that controls the effective step size and the speed of convergence. In this simple implementation, the level of damping is a control parameter of the algorithm. A more sophisticated approach would use adaptive damping $\lambda_t$, where the effective step size decreases as the Bethe free energy of the configuration $\{\boldsymbol{x}_\alpha^t\}$ decreases [7, 3].

## 4    Noncubic tensors

In the above, we have assumed that the dimensionality of all $p$ modes is $N$, implying that the underlying tensor is cubic (i.e., all modes have the same length). To study how the shape of the tensors influence the AMP algorithm and the performance, we allow for the different modes to have different dimensionality $N_\alpha$. Importantly, we assume that all modes are in the thermodynamic

regime, i.e., $N_\alpha \to \infty$ $\alpha = \{1, ..., p\}$. Furthermore, we assume all modes scale in a similar way. This is done by defining $N_\alpha = n_\alpha N$ where all $n_\alpha = O(1)$ and $\prod_\alpha n_\alpha = 1$. The thermodynamic limit is then understood by taking $N \to \infty$.

First we note that the scaling of the tensor elements does not change with this choice of scaling,

$$w_b \sim \sqrt{\frac{N}{\prod_\alpha N_\alpha}} \sim \frac{N^{-\frac{p-1}{2}}}{\sqrt{\prod_\alpha n_\alpha}} = N^{\frac{1-p}{2}}.$$

However, the algorithms have no symmetry with respect to the dimensionality of the different modes in this case. This broken symmetry is in the iterative mean-field equations for the local mean messages $\mathbf{u}_{\alpha i}$, in eq. (43), which now is scaled by proportion of the dimensionality respective mode:

$$\mathbf{u}_{\alpha i}^t = \frac{n_\alpha}{N^{(p-1)/2}} \sum_{b \in \partial \alpha i} S_b \prod_{\beta j \in \partial b \setminus \alpha i}^{\odot} \hat{\mathbf{x}}_{\beta j}^t - \frac{1}{\Delta} \hat{\mathbf{x}}_{\alpha i}^{t-1} \sum_{\beta \neq \alpha} \Sigma_\beta^t \odot D_{\alpha \beta}^t \tag{72}$$

The other mean-field equations of the algorithms are left unchanged.

**Correction to the dynamic mean-filed equations** In order to make the necessary changes to the dynamic mean-field theory in section 2, we redefine the mean overlap with the appropriate scaling, which now depends on the mode $\alpha$,

$$M_\alpha^t = \frac{1}{n_\alpha N} \sum_i^N \hat{\mathbf{x}}_{\alpha i}^t \mathbf{x}_{\alpha i}^{0T} \in \mathbb{R}^{r \times r}. \tag{73}$$

Using the rescaled overlap, we re-derive the iterative dynamic mean-field equations, following the same steps as in section 2:

$$\mathbb{E}\left[\mathbf{u}_{\alpha i}^t\right] = \frac{n_\alpha}{N^{(p-1)/2} \sqrt{\prod_\beta n_\beta}} \times$$

$$\sum_{b \in \partial \alpha i} P_{out}(Y_b, 0) w_b \left(\frac{\partial \log P_{out}(Y_b, w)}{\partial w}\right)_{w=0}^2 \prod_{\beta j \in \partial b \setminus \alpha i}^{\odot} \hat{\mathbf{x}}_{\beta j \to b}^t =$$

$$\frac{1}{\Delta N^{(p-1)/2}} \sum_{b \in \partial \alpha i} w_b \prod_{\beta j \in \partial b \setminus \alpha i}^{\odot} \hat{\mathbf{x}}_{\beta j \to b}^t. \tag{74}$$

Substituting the expression for $w$,

$$\mathbb{E}\left[\mathbf{u}_{\alpha i}^t\right] = \frac{n_\alpha}{\Delta N^{(p-1)} \prod_\beta n_\beta} \sum_{b \in \partial \alpha i} \prod_{(\beta j) \in \partial b}^{\odot} \mathbf{x}_{\beta j}^0 \prod_{\beta j \in \partial b \setminus \alpha i}^{\odot} \hat{x}_{\beta j}^{tT}$$

$$= \frac{n_\alpha}{\Delta N^{(p-1)} \prod_\beta n_\beta} \sum_{b \in \partial \alpha i} \left(\mathbf{x}_{\alpha i}^0\right)^T \left(\prod_{\beta j \in \partial b \setminus \alpha i}^{\odot} \mathbf{x}_{\beta j}^0\right) \left(\prod_{\beta j \in \partial b \setminus \alpha i}^{\odot} \hat{\mathbf{x}}_{\beta j}^t\right)^T$$

$$= \frac{n_\alpha}{\Delta} \left(\mathbf{x}_{\alpha i}^0\right)^T \prod_{\beta \neq \alpha}^{\odot} \left(\frac{1}{N n_\beta} \sum_j^N \mathbf{x}_{\beta j}^0 \hat{\mathbf{x}}_{\beta j}^{tT}\right) = \frac{n_\alpha}{\Delta} \left(\mathbf{x}_{\alpha i}^0\right)^T \prod_{\beta \neq \alpha}^{\odot} M_\beta^t \tag{75}$$

Finally we arrive at

$$\mathbb{E}\left[\mathbf{u}_{\alpha i}^t\right] = \frac{n_\alpha}{\Delta} \prod_{\beta \neq \alpha}^{\odot} M_\beta^t \mathbf{x}_{\alpha i}^0. \tag{76}$$

Note that the only difference between this result and eq. (58) is the factor $n_\alpha$. It follows that in order to generalize the dynamic mean-field theory to noncubic tensors, we simply need to replace $M_\alpha \to n_\alpha M_\alpha$ throughout the results of section 2. The final equations are presented in the main text.

# 5 Solutions to the dynamic mean-field equations with specific priors

In order to theoretically evaluate the performance of the AMP algorithms given different tensors, we explicitly derive the dynamic mean-field equation of the overlap, and the error, for some common priors $P_\alpha(x)$. These derivations closely resemble the analysis done in [3] for the $p = 2$ case, only here we allow different mixture of prior and different sizes for the modes, and consider arbitrary order $p$. In the following we will use rank $r = 1$ tensors, where the estimators $x$, the ground truth $x^0$, and the the overlaps $M$, which we will denote here as $m$, are all scalar values. The same analysis hods with multivariate calculation, when $r \geq 2$.

In order to theoretically evaluate the performance of the AMP algorithms given different tensors, we explicitly derive the dynamic mean-field equations of the overlap, and the error, for some common priors $P_\alpha(x)$. These derivations closely resemble the analysis done in [3] for the $p = 2$ case, only here we allow a different mixture of prior and different sizes for the modes, and consider arbitrary order $p$. In the following, we will use rank $r = 1$ tensors, where the estimators $x$, the ground truth $x^0$, and the overlaps $M$ – which we will denote here as m – are all scalar values. The same analysis holds with multivariate calculation when $r \geq 2$.

## 5.1 Gaussian prior

The first, and perhaps most common, choice for prior is a normal distribution of $x_\alpha$, with variance $\sigma_\alpha^2$ and and mean $\mu_\alpha$,

$$P_\alpha(x) = \frac{1}{\sqrt{2\pi}\sigma_\alpha}e^{-(x-\mu_\alpha)^2/2\sigma_\alpha^2}. \tag{77}$$

We use that prior to explicitly calculate the update rule,

$$f_\alpha = \frac{\partial}{\partial \mathbf{u}}\mathcal{Z}_\alpha(A, \mathbf{u}) = \frac{\int d\mathbf{x}P_\alpha(\mathbf{x})\mathbf{x}e^{-\frac{1}{2}\mathbf{x}^T A\mathbf{x}+\mathbf{u}^T\mathbf{x}}}{\int d\mathbf{x}P_\alpha(\mathbf{x})e^{-\frac{1}{2}\mathbf{x}^T A\mathbf{x}+\mathbf{u}^T\mathbf{x}}}. \tag{78}$$

The nominator of (78) can be written as

$$\int dx x \frac{1}{\sqrt{2\pi}}e^{-(x-\mu_\alpha)^2/2\sigma_\alpha^2}e^{-\frac{1}{2}Ax^2+ux}$$

$$= \int dx x \frac{1}{\sqrt{2\pi}\sigma_\alpha}\exp\frac{1}{2\sigma_\alpha^2}\left[-x^2 + 2x\mu_\alpha - \mu_\alpha^2 - ax + 2bx\right]$$

$$= \int dx x \frac{1}{\sqrt{2\pi}\sigma_\alpha}\exp\frac{-1}{2\sigma_\alpha^2}\left[(a+1)x^2 - 2(\mu_\alpha + b)x + \mu_\alpha^2\right], \tag{79}$$

where $b = u\sigma_\alpha^2$ and $a = A\sigma_\alpha^2$. Completing the quadratic form, we have

$$= \int dx x \frac{1}{\sqrt{2\pi}\sigma_\alpha}\exp\frac{-1}{2\sigma_\alpha^2}\left[\left(\sqrt{a+1}x - \frac{\mu_\alpha + b}{\sqrt{a+1}}\right)^2 + \mu_\alpha^2 - \frac{(\mu_\alpha + b)^2}{a+1}\right]$$

$$= \frac{1}{\sqrt{a+1}}\exp\left[\frac{-1}{\sigma_\alpha^2}\mu_\alpha^2 - \frac{(\mu_\alpha + b)^2}{a+1}\right]\int dx x \frac{\sqrt{a+1}}{\sqrt{2\pi}\sigma_\alpha}\exp\frac{-(a+1)}{\sigma_\alpha^2}\left[\left(x - \frac{\mu_\alpha + b}{a+1}\right)^2\right].$$

Similar treatment in performed on the denominator. It is straight forward to see that the function in (78) reduces to

$$f_\alpha(A, u) = \frac{\mu_\alpha + u\sigma_\alpha^2}{A\sigma_\alpha^2 + 1}. \tag{80}$$

Next, we want to use this functional form in the dynamic mean-field eq. (66). Denote $\tilde{m}_\alpha^t = \frac{1}{\Delta}\prod_{\beta\neq\alpha}^{\odot}m_\beta^t$, then we have $A^t = \hat{m}_\alpha^t$ and $\mathbf{u}_\alpha^t = \hat{m}_\alpha^t x_\alpha^0 + \sqrt{\hat{m}_\alpha^t}z$ then we want to compute

$$\left\langle \frac{\frac{\mu_\alpha}{\sigma_\alpha^2} + \mathbf{u}}{A + \frac{1}{\sigma_\alpha^2}}x^0\right\rangle_{z,x_\alpha^0} = \left\langle \frac{\frac{\mu_\alpha}{\sigma_\alpha^2} + \hat{m}_\alpha^t x_\alpha^0 + \sqrt{\tilde{m}_\alpha^t}z}{\tilde{m}_\alpha^t + \frac{1}{\sigma_\alpha^2}}x^0\right\rangle_{z,x_\alpha^0} = \left\langle \frac{\frac{\mu_\alpha}{\sigma_\alpha^2} + \tilde{m}_\alpha^t x_\alpha^0}{\tilde{m}_\alpha^t + \frac{1}{\sigma_\alpha^2}}x^0\right\rangle_{x^0}, \tag{81}$$

Averaging over the distribution of the ground-truth values $P(x^0)$,

$$\left\langle f_\alpha \left(\tilde{m}_\alpha^t\right) x^0 \right\rangle_{P_\alpha} = \frac{\frac{\mu_\alpha^2}{\sigma_\alpha^2} + \tilde{m}_\alpha^t \left(\sigma_\alpha^2 + \mu_\alpha^2\right)}{\tilde{m}_\alpha^t + \frac{1}{\sigma_\alpha^2}}. \tag{82}$$

Finally the dynamic mean-field iterative equation on the mean overlap are given by

$$m_\alpha^{t+1} = \frac{\Delta \frac{\mu_\alpha^2}{\sigma_\alpha^2} + \left(\sigma_\alpha^2 + \mu_\alpha^2\right) \prod_{\beta \neq \alpha}^{\odot} m_\beta^t}{\frac{\Delta}{\sigma_\alpha^2} + \prod_{\beta \neq \alpha}^{\odot} m_\beta^t}. \tag{83}$$

In the case of zero mean priors, $\mu_\alpha = 0$, the equation is reduced to

$$m_\alpha^{t+1} = \frac{\sigma_\alpha^2 \prod_{\beta \neq \alpha}^{\odot} m_\beta^t}{\frac{\Delta}{\sigma_\alpha^2} + \prod_{\beta \neq \alpha}^{\odot} m_\beta^t}. \tag{84}$$

We note that if all modes are Gaussian with zero means, then the solution $M_\alpha = 0 \ \forall \alpha$ is a stable fixed point of the dynamics, implying that if we start from random initial conditions, that are uncorrelated with the true values, the algorithms will not converge. A numerical analysis of eq. (83) for order $p = 3$ tensors is presented in the main text.

Consider the case of $\mu_\alpha = \mu$ and $\sigma_\alpha = \sigma$, with all priors are similar. From the structure of (83) we find that in the fixed point $M_\alpha^*$

$$m_\alpha^* = m^* \ \forall \alpha, \tag{85}$$

which is what would be expected from the symmetry of the problem. Note however that unlike the derivation in [4], the underlying tensor in non-symmetric.

**Noncubic tensors**. If we have different population sizes, then we have a ratio between the order parameters, and replace $m_\alpha$ with $n_\alpha m_\alpha$.

$$m_\alpha^{t+1} = \frac{\Delta \frac{\mu_\alpha^2}{\sigma_\alpha^2} + \left(\sigma_\alpha^2 + \mu_\alpha^2\right) \prod_{\beta \neq \alpha} n_\beta m_\beta^t}{\frac{\Delta}{\sigma_\alpha^2} + \prod_{\beta \neq \alpha} k_\beta m_\beta^t} \tag{86}$$

## 5.2 Bernoulli distribution

For many applications, it is expected that some of the modes in the underlying low rank tensors are sparse, meaning they contribute information to only a small subset of the measurements. A simple way of modeling such data is using the Bernoulli distribution,

$$P_\alpha(x) = \rho \delta(x - 1) + (1 - \rho)\delta(x). \tag{87}$$

As in the derivation of the Gaussian priors in the previous subsection, we compute the function (78). The nominator is equal to

$$\int dx \left[\rho \delta(x - 1) + (1 - \rho)\delta(x)\right] x e^{-\frac{1}{2}x^T A x + \mathbf{u}^T x} =$$

$$\rho e^{-\frac{1}{2}\sum_{ij} A + \sum_j \mathbf{u_j}}, \tag{88}$$

and the denominator is given by

$$\int dx \left[\rho \delta(x - 1) + (1 - \rho)\delta(x)\right] e^{-\frac{1}{2}x^T A x + \mathbf{u}^T x} =$$

$$\rho e^{-\frac{1}{2}\sum_{ij} A + \sum_j \mathbf{u}} + (1 - \rho). \tag{89}$$

Combining both expressions together we get

$$f_\alpha(A, u) = \frac{\rho e^{-\frac{1}{2}A + \mathbf{u}}}{\rho e^{-\frac{1}{2}A + \mathbf{u}} + (1 - \rho)} = \frac{\rho}{\rho + (1 - \rho)e^{\frac{1}{2}A - \mathbf{u}}}, \tag{90}$$

with first derivative equal to

$$\frac{\partial}{\partial \mathbf{u}} f_\alpha = \frac{e^{-\frac{1}{2}A+\mathbf{u}}\left(\rho^{-1}-1\right)}{\left[\left(e^{-\frac{1}{2}A+\mathbf{u}}-1\right)+\rho^{-1}\right]^2}. \tag{91}$$

In the bayes optimal case we have $A_\alpha = \tilde{m}_\alpha$ and $u_{\alpha i} = \tilde{m}_\alpha x_i^0 + \sqrt{\tilde{m}_\alpha} z$ ,where we have defined $\tilde{m}_\alpha \equiv \frac{1}{\Delta} \prod_{\beta \neq \alpha} m_\beta$. In the expression in the exponent of (91) we have

$$\frac{1}{2}A - u_i = \tilde{m}_\alpha \left(\frac{1}{2} - x_i^0\right) + \sqrt{\tilde{m}_\alpha} z. \tag{92}$$

Next we integrate over the prior and ground-truth to get

$$m_\alpha^{t+1} = \rho \mathbb{E}_z \left[ f_\alpha \left( \tilde{m}_\alpha, \tilde{m}_\alpha + \sqrt{\tilde{m}_\alpha} z \right) \right] =$$
$$\rho^2 \left\langle \left( \rho + (1-\rho)\exp\left[\frac{1}{2}\tilde{m}_\alpha - \sqrt{\tilde{m}_\alpha} z\right] \right)^{-1} \right\rangle_z \tag{93}$$

In the sparse case, where $\rho \ll 1$ this can be simplified further

$$m_\alpha^{t+1} = \rho^2 \left\langle \left( \exp\left[-\frac{1}{2}\tilde{m}_\alpha + \sqrt{\tilde{m}_\alpha} z\right] \right) \right\rangle_z + \mathcal{O}(\rho^3)$$
$$= \frac{\rho^2}{\sqrt{2\pi}} \int_{-\infty}^{\infty} dz \exp\left[-\frac{z^2}{2} - \frac{1}{2}\tilde{m}_\alpha + \sqrt{\tilde{m}_\alpha} z\right] + \mathcal{O}(\rho^3)$$
$$= \rho^2 e^{\tilde{m}_\alpha/2} + \mathcal{O}(\rho^3) \tag{94}$$

Note that in a complete overlap we have $m = \rho$ so $e^{\tilde{m}_\alpha/2} = 1/\rho$ and

$$\frac{1}{2\Delta} \prod_{\beta \neq \alpha} m_\beta = -\log \rho$$

In instances where all of the modes have similar statistics, then we would have

$$\frac{1}{2\Delta}\rho^{p-1} = -\log \rho \Rightarrow \Delta = \frac{\rho^{p-1}}{2\left|\log \rho\right|}.$$

Here, we can expect that for $\Delta \sim \rho^{p-1}/\left|\log \rho\right|$, where $p$ is the order of the tensor, we will have high overlap with zero error. However, in the case of non-symmetric tensors, not all directions have to be sparse, and may have different distributions. In that case the noise scale as $\Delta \sim \rho^{\tilde{p}-1}/\left|\log \rho\right|$, where $\tilde{p}$ is the number of sparse modes in the underlying tensor.

### 5.3 Gauss-Bernoulli

The next logical step is to combine the continuous irregularity of the Gaussian distribution and the sparse nature of the Bernoulli distribution. The Gauss-Bernoulli distribution is given by

$$P_\alpha(x) = \rho \mathcal{N}(\mu, \sigma^2) + (1-\rho)\delta(x). \tag{95}$$

For brevity we will use zero mean $\mu = 0$ and unit variance $\sigma^2 = 1$, and note that the results can be easily rescaled. The update function is given by

$$f_\alpha(A, u) = \frac{\rho \int dx x \frac{1}{\sqrt{2\pi}} e^{-\frac{1}{2}x^T A x + \mathbf{u}^T x - \frac{1}{2}x^2}}{\rho \int dx \frac{1}{\sqrt{2\pi}} e^{-\frac{1}{2}x^T A x + \mathbf{u}^T x - \frac{1}{2}x^2} + (1-\rho)}.$$

Using some algebra we get

$$f_\alpha = \frac{\rho \int dx\, x \frac{1}{\sqrt{2\pi}} \exp\left[-\frac{1}{2}(\sqrt{A-1}x - \frac{u}{\sqrt{A+1}})^2 + \frac{u^2}{(A+1)}\right]}{\rho \int dx \frac{1}{\sqrt{2\pi}} \exp\left[-\frac{1}{2}(\sqrt{A-1}x - \frac{u}{\sqrt{A+1}})^2 + \frac{u^2}{(A+1)}\right] + (1-\rho)} =$$

$$\frac{\frac{1}{\sqrt{A+1}}\rho \int dx\, x \frac{\sqrt{A+1}}{\sqrt{2\pi}} \exp\left[-\frac{\sqrt{A-1}}{2}(x - \frac{u}{A+1})^2\right] e^{\frac{u^2}{(A+1)}}}{\frac{1}{\sqrt{A+1}}\rho \int dx \frac{\sqrt{A+1}}{\sqrt{2\pi}} \exp\left[-\frac{\sqrt{A-1}}{2}(x - \frac{u}{A+1})^2\right] e^{\frac{u^2}{(A+1)}} + (1-\rho)} =$$

$$\frac{\rho u}{(A+1)\rho + (1-\rho)(A+1)^{3/2} e^{\frac{-u^2}{(A+1)}}} \quad (96)$$

and the first derivative is given by

$$\frac{\partial f_\alpha}{\partial u} = \rho \frac{\left((A+1)\rho + (1-\rho)(A+1)^{3/2} e^{\frac{-u^2}{(A+1)}}\right) + 2(1-\rho)u^2 (A+1)^{1/2} e^{\frac{-u^2}{(A+1)}}}{\left((A+1)\rho + (1-\rho)(A+1)^{3/2} e^{\frac{-u^2}{(A+1)}}\right)^2}. \quad (97)$$

For sanity check, if $\rho = 1$ then

$$f_\alpha(\rho = 1) = \frac{u}{A+1}$$

$$\frac{\partial f_\alpha(\rho = 1)}{\partial u} = \frac{1}{A+1}$$

and we have recovered the results for the Gaussian priors from above. From here we can calculate the dynamic mean-filed equations

$$m_\alpha^{t+1} = \rho \int P_\alpha(x^0) dx^0 \frac{dz}{\sqrt{2\pi}} e^{-z^2/2} \frac{\tilde{m}_\alpha^t x^0 + \sqrt{\frac{1}{\Delta} \tilde{m}_\alpha^t} z}{(\tilde{m}_\alpha^t + 1)\rho + (1-\rho)(\tilde{m}_\alpha^t + 1)^{3/2} \exp\left[-\frac{\left(\tilde{m}_\alpha^t x^0 + \sqrt{\tilde{m}_\alpha^t} z\right)^2}{\tilde{m}_\alpha^t + 1}\right]} x^0 \quad (98)$$

$$= \rho^2 \frac{\tilde{m}_\alpha^t}{(\tilde{m}_\alpha^t + 1)} \int \frac{dz\, dx^0}{2\pi} \exp\left(-\frac{x^{02} + z^2}{2}\right) \frac{\left(x^0\right)^2}{\rho + (1-\rho)(\tilde{m}_\alpha^t + 1)^{1/2} \exp\left[-\frac{\left(\tilde{m}_\alpha^t x^0 + \sqrt{\tilde{m}_\alpha^t} z\right)^2}{\tilde{m}_\alpha^t + 1}\right]}$$

## 5.4 Mixed priors

In the case of general asymmetric tensors, we can construct a tensor using different priors for the different modes. It is particularly useful in real applications, as different modes of the tensors can originate from entirely different sources. Consider for example an order-3 tensor holding neural firing rate data $r_{itk}$. The index $i$ marks the neuron recorded; index $t$ is the time bin within a single trial, and $k$ is the trial index. If we believe that the data originates from the low-dimensional dynamical system, we would want to write the tensor as

$$r_{itk} = \sum_\rho^D u_i^\rho x_t^\rho v_k^\rho + \sqrt{\Delta}\epsilon_{itk}, \quad (99)$$

where $D$ is the dimensions of the dynamical system, and $\Delta$ is the noise of a single measurement. We may ask how should we design an experiment so that low-rank decomposition of the recorded data would be possible. In this case, we would assert different priors to the different modes. The mode $x_t$ represent the $D$ dimensional dynamical system. We could assume for example that is generated by some Gaussian process, thus follows Gaussian statistics. The mode $u_i$ represents the projections of the low-dimensional dynamical system onto the set measured neurons. It may be a valid

assumption that only a fraction of the neurons responds in coherence with the underlying dynamics; a Gauss-Bernoulli distribution will be suitable for this mode. Lastly, the trial modulus mode $v_k$ can have Gaussian distribution about some mean with small variance, suggesting small trial-to-trial modulations.

To solve the dynamic mean field theory for this case, and find the boundaries of the inference we would use the appropriate equation for each of the modes. For example, for two Gaussian distributions and one Gauss-Bernoulli, we would have

$$m_x^{t+1} = \frac{\Delta \frac{\mu_x^2}{\sigma_x^2} + \left(\sigma_x^2 + \mu_x^2\right) m_u^t m_v^t}{\frac{\Delta}{\sigma_\alpha^2} + m_u^t m_v^t} \tag{100}$$

$$m_v^{t+1} = \frac{\Delta \frac{\mu_v^2}{\sigma_v^2} + \left(\sigma_v^2 + \mu_v^2\right) m_u^t m_x^t}{\frac{\Delta}{\sigma_\alpha^2} + m_u^t m_x^t} \tag{101}$$

$$m_u^{t+1} = \rho^2 \frac{\tilde{m}_\alpha^t}{\left(\tilde{m}_\alpha^t + 1\right)} \int \frac{dz\, dx^0}{2\pi} \exp\left(-\frac{x^{02} + z^2}{2}\right) \tag{102}$$

$$\times \frac{\left(x^0\right)^2}{\rho + (1-\rho)\left(\frac{1}{\Delta} m_x^t m_v^t + 1\right)^{1/2} \exp\left[-\frac{\left(\frac{1}{\Delta} m_x^t m_v^t x^0 + \sqrt{\frac{1}{\Delta} m_x^t m_v^t} z\right)^2}{\frac{1}{\Delta} m_x^t m_v^t + 1}\right]} \tag{103}$$

This set of equations can be solved numerically, to find an estimate for AMP performances under the noise.