[Reviews · NeurIPS 2018]

Reviewer 1



The algebraic problem of low rank tensor decomposition still does not have a satisfactory solution. Even finding the tensor rank is NP-hard [arXiv:0911.1393]. The paper assumes in eq(1) that the rank r is known a-priory which is rarely the case. But under this assumption the paper proceeds to systematically derive Approximate Message Passing algorithm in the large-N limit where mean-field approximation becomes exact. The paper discovers several inference regimes depending on the signal-to-noise levels (SNR). At SNR below curtain threshold inference becomes impossible. Detailed comparisons of AMP to Alternating Least Squares (ALS) are provided as well. The paper is reasonably well written. However, it lacks any practical example problem where a low-rank tensor is embedded in additive Gaussian noise. Only synthetic numerical examples are provided.

Reviewer 2



Summary: Aim of the paper is to provide a better theoretical understanding of the approximate message passing (AMP) algorithm for low-rank tensor decomposition. For this method, the author show that a mean field approximation may be used to reveal the existence of different regimes (phases) where the inference is easy, hard or impossible. They also extend the AMP algorithm of previous works to the case of non-symmetric tensors. Strengths: Low-rank tensor decomposition is a widespread and hard problem. Defining regimes where inference is possible/impossible may help obtain one a better high-level interpretation of the task. The analysis proposed by the authors is theoretically sound and connects the tensor factorization task with nice concepts from the physics literature. Such connections have been shown to be useful and inspiring in many other cases in computer science. Finally, the proposed algorithm considerably extends the applicability of the AMP approach. Weaknesses: It looks complicated to assess the practical impact of the paper. On the one hand, the thermodynamic limit and the Gaussianity assumption may be hard to check in practice and it is not straightforward to extrapolate what happens in the finite dimensional case. The idea of identifying the problem's phase transitions is conceptually clear but it is not explicitly specified in the paper how this can help the practitioner. The paper only compares the AMP approach to alternate least squares without mention, for example, positive results obtained in the spectral method literature. Finally, it is not easy to understand if the obtained results only regard the AMP method or generalize to any inference method. Questions: - Is the analysis restricted to the AMP inference? In other words, could a tensor that is hard to infer via AMP approach be easily identifiable by other methods (or the other way round)? - Are the easy-hard-impossible phases be related with conditions on the rank of the tensor? - In the introduction the authors mention the fact that tensor decomposition is in general harder in the symmetric than in the non-symmetric case. How is this connected with recent findings about the `nice' landscape of the objective function associated with the decomposition of symmetric (orthogonal) order-4 tensors [1]? - The Gaussian assumption looks crucial for the analysis and seems to be guaranteed in the limit r << N. Is this a typical situation in practice? Is always possible to compute the `effective' variance for non-gaussian outputs? Is there a finite-N expansion that characterize the departure from Gaussianity in the non-ideal case? - For the themodynamic limit to hold, should one require N_alpha / N = O(1) for all alpha? - Given an observed tensor, is it possible to determine the particular phase it belongs to? [1] Rong Ge and Tengyu Ma, 2017, On the Optimization Landscape of Tensor Decompositions

Reviewer 3



This paper derive a Bayesian approximate message passing (AMP) algorithms for recovering arbitrarily shaped low-rank tensors with additive noise and employ dynamic mean field theory to precisely characterize their performance. The theory reveals the existence of phase transitions between easy, hard and impossible inference regimes. The main contribution is to solve the low-rank tensor decomposition by AMP algorithm together with some theoretical properties. The clarity of the paper need to be further improved. Strength: This paper developed a new algorithm to solve low-rank tensor decomposition which shows much better than standard ALS algorithm. The theoretical analysis of this work is interesting. Weakness: Since approximate message passing algorithm is well know, and low-rank tensor decomposition is formulated as a very standard probabilistic model with Gaussian noise, thus it is straightforward to apply AMP for low-rank decomposition. The novelty is incremental. The experiment part is very weak and insufficient. It is only compared with very simple ALS algorithm. There are lots of algorithms developed for low-rank tensor decomposition including probabilistic tensor decomposition and Bayesian tensor decomposition. The comparison is far less than sufficient. There is only one simple experiment on synthetic data, the data size is also quite small with N=500. The experiment on real-world dataset or applications is missing, which is not convincing and promising from practical point of view.

Reviewer 4



This line of research seems interesting, as it proposes an alternative view on the computation of low-rank tensor decomposition. On the other hand, my impression is that the paper itself needs some more work as it is quite technical and difficult to read. There are also some assumptions that should be better justified, such as r\infinity (last paragraph of Section 2.1). The simulations are useful but somewhat limited. In particular, it is unclear what kind of practical applications (if any) the authors have in mind.